# Optimization or Architecture:
# How to Hack Kalman Filtering

**Ido Greenberg**
Technion
gido@campus.technion.ac.il

**Netanel Yannay**
ELTA Systems
natiy4@gmail.com

**Shie Mannor**
Technion, Nvidia Research
shie@ee.technion.ac.il

## Abstract

In non-linear filtering, it is traditional to compare non-linear architectures such as neural networks to the standard linear Kalman Filter (KF). We observe that this mixes the evaluation of two separate components: the non-linear architecture, and the parameters optimization method. In particular, the non-linear model is often optimized, whereas the reference KF model is not. We argue that *both* should be optimized similarly, and to that end present the Optimized KF (**OKF**). We demonstrate that the KF may become competitive to neural models – if optimized using OKF. This implies that experimental conclusions of certain previous studies were derived from a flawed process. The advantage of OKF over the standard KF is further studied theoretically and empirically, in a variety of problems. Conveniently, OKF can replace the KF in real-world systems by merely updating the parameters. Our experiments are published in Github, and the OKF in PyPI.

## 1 Introduction

The Kalman Filter (KF) [Kalman, 1960] is a celebrated method for linear filtering and prediction, with applications in many fields including tracking, navigation, control and reinforcement learning [Zarchan and Musoff, 2000, Kirubarajan, 2002, Kaufmann et al., 2023]. The KF provides optimal predictions under certain assumptions (namely, linear models with i.i.d noise). In practical problems, these assumptions are often violated, rendering the KF sub-optimal and motivating the growing field of non-linear filtering. Many studies demonstrated the benefits of non-linear models over the KF [Revach et al., 2022, Coskun et al., 2017].

Originally, we sought to join this line of works. Motivated by a real-world Doppler radar problem, we developed a dedicated non-linear Neural KF (NKF) based on the LSTM sequential model. NKF achieved significantly better accuracy than the linear KF.

Then, during ablation tests, we noticed that the KF and NKF differ in *both architecture and optimization*. Specifically, the KF's noise parameters are traditionally determined by noise estimation [Odelson et al., 2006]; whereas NKF's parameters are optimized using supervised learning. To fairly evaluate the two architectures, we wished to apply the same optimization to both. To that end, we devised an Optimized KF (**OKF**, Section 3). KF and OKF have the same linear architecture: OKF only changes the noise parameters values. Yet, unlike KF, OKF *outperformed* NKF, which reversed the whole experimental conclusion, and made the neural network unnecessary for this problem (Section 4).

Our original error was comparing two different model architectures (KF and NKF) that were not optimized similarly. A review of the non-linear filtering literature reveals that this methodology is used in many studies. Specifically, for a baseline KF model, the parameters are often tuned by noise estimation [fa Dai et al., 2020, Aydogmus and Aydogmus, 2015, Revach et al., 2022]; by heuristics [Jamil et al., 2020, Coskun et al., 2017, Ullah et al., 2019]; or are simply ignored [Gao et al., 2019, Bai et al., 2020, Zheng et al., 2019], often without public code for examination. Hussein [2014]

37th Conference on Neural Information Processing Systems (NeurIPS 2023).

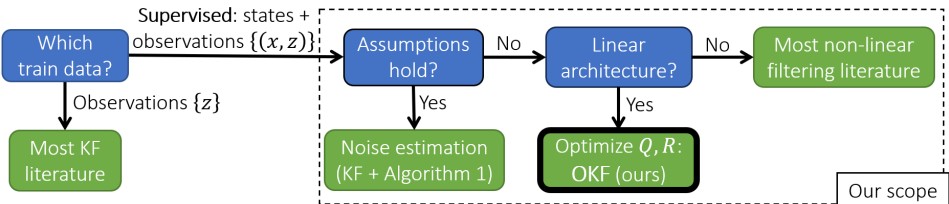

Figure 1: Since the KF assumptions are often violated, noise estimation does not optimize the MSE. Instead, our method (OKF) optimizes the MSE directly. In particular, neural network models should be tested against OKF rather than the non-optimized KF – in contrast to the common practice in the literature.

even discusses the (Extended-)KF sensitivity to its parameters, and suggests a neural network with supervised learning – yet never considers the same supervised learning for the KF itself. In all these 10 studies, the non-linear model's added value cannot be inferred from the experimental evidence.

So far, OKF is presented as a *methodological contribution*: for comparison with non-linear methods, it forms a more coherent baseline than KF. In addition, OKF provides a *practical contribution*: it achieves more accurate filtering than the KF, using identical architecture. This is demonstrated extensively in Section 5 and Appendix B – in different domains, over different problem variations, using different KF baselines, with different data sizes, and even under distributional shifts.

**Discrepancy of objectives**: The advantage of OKF over KF may come as a surprise: KF's standard noise estimation is known to already obtain the MSE-optimal parameters! However, this optimality relies on unrealistic assumptions, often considered "fairy tales for undergraduates" [Thomson, 1994]. When violated, a conflict emerges between noise estimation and MSE optimization. We study this in detail: (a) Section 5 analyzes the conflict theoretically under certain assumption violations; (b) Appendix B.1 shows that even oracle noise estimation cannot optimize the MSE; (c) Appendix B.2 shows that when using noise estimation, **the MSE may degrade with more data**. In this light, all our findings can be summarized as follows (also see Fig. 1):

**Contribution:** (a) We observe that in most scenarios, since the KF assumptions do not hold, noise estimation is *not* a proxy to MSE optimization. (b) We thus present the Optimized KF (OKF), also available as a PyPI package. We analyze (theoretically and empirically) the consequences of neglecting to optimize the KF: (c) the standard KF tuning method leads to sub-optimal predictions; (d) the standard methodology in the literature compares an optimized model to a non-optimized KF, hence may produce misleading conclusions. Note that we *do not* argue against the non-linear models; rather, we claim that their added value cannot be deduced from flawed experiments.

**Scope: We focus on the supervised filtering setting**, where training data includes both observations and the true system states (whose prediction is usually the objective). Such data is available in many practical applications. For example, the states may be provided by external accurate sensors such as GPS fa Dai et al. [2020]; by manual object labeling in computer vision [Wojke et al., 2017]; by controlled experiments of radar targets; or by simulations of known dynamic systems.

As demonstrated in the 10 studies cited above, this supervised setting is common in non-linear filtering. In linear Kalman filtering, this setting seems to be solved by trivial noise estimation; thus, the literature tends to overlook it, and instead focuses on settings that do not permit trivial noise estimation, e.g., learning from observations alone. Nevertheless, we argue that even in the supervised setting, noise estimation is often not a proxy to MSE optimization, and thus should often be avoided.

## 2   Preliminaries

Consider the KF model for a dynamic system with no control signal [Kalman, 1960]:

$$X_{t+1} = F_t X_t + \omega_t \quad (\omega_t \sim \mathcal{N}(0, Q)), \qquad Z_t = H_t X_t + \nu_t \quad (\nu_t \sim \mathcal{N}(0, R)). \qquad (1)$$

$X_t$ is the system state at time $t$, and its estimation is usually the goal. Its dynamics are modeled by the linear operator $F_t$, with random noise $\omega_t$ whose covariance is $Q$. $Z_t$ is the observation, modeled by the operator $H_t$ with noise $\nu_t$ whose covariance is $R$. The notation may be simplified to $F, H$ in the stationary case.

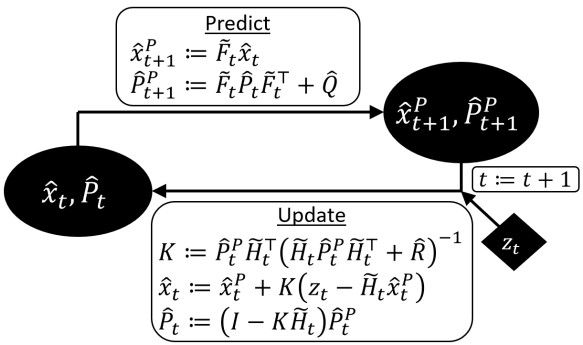

Figure 2: The KF algorithm. The prediction step is based on the motion model $\tilde{F}_t$ with noise $\hat{Q}$, whereas the update step is based on the observation model $\tilde{H}_t$ with noise $\hat{R}$.

The KF represents $X_t$ via estimation of the mean $\hat{x}_t$ and covariance $\hat{P}_t$. As shown in Fig. 2, the KF alternately predicts the next state (*prediction* step), and processes new information from incoming observations (*update* or *filtering* step). The KF relies on the matrices $\tilde{F}_t, \tilde{H}_t, \hat{Q}, \hat{R}$, intended to represent $F_t, H_t, Q, R$ of Eq. (1). Whenever $F_t, H_t$ are known and stationary, we may simplify the notation to $\tilde{F}_t = F$, $\tilde{H}_t = H$.

The KF estimator $\hat{x}_t$ is optimal in terms of mean square errors (MSE) – but only under a restrictive set of assumptions [Kalman, 1960]:

**Assumption 1** (KF assumptions). $\tilde{F}_t = F_t$, $\tilde{H}_t = H_t$ are known and independent of $X_t$ (linear models); each sequence $\{\omega_t\}, \{\nu_t\}$ is i.i.d; the covariances $\hat{Q} = Q$, $\hat{R} = R$ are known; and $\hat{x}_0, \hat{P}_0$ correspond to the mean and covariance of the initial $X_0$.

**Theorem 1** (KF optimality; e.g., see Jazwinski [2007], Humpherys et al. [2012]). Under Assumption 1, the KF estimator $\hat{x}_t$ minimizes the MSE w.r.t. $X_t$.

The KF accuracy strongly depends on its parameters $\hat{Q}$ and $\hat{R}$ [Formentin and Bittanti, 2014]. As motivated by Theorem 1, these parameters are usually identified with the noise covariance $Q, R$ and are set accordingly: "the systematic and preferable approach to determine the filter gain is to estimate the covariances from data" [Odelson et al., 2006]. In absence of system state data $\{x_t\}$ (the "ground truth"), many methods were suggested to estimate the covariances from observations $\{z_t\}$ alone [Mehra, 1970, Zanni et al., 2017, Park et al., 2019, Feng et al., 2014]. We focus on the supervised setting, where the states $\{x_t\}$ are available in the training-data (but not in inference).

**Definition 1** (Supervised data). Consider $K$ trajectories of a dynamic system, with lengths $\{T_k\}_{k=1}^K$. We define their supervised data as the sequences of true system states $x_{k,t} \in \mathbb{R}^{d_x}$ and observations $z_{k,t} \in \mathbb{R}^{d_z}$: $\{\{(x_{k,t}, z_{k,t})\}_{t=1}^{T_k}\}_{k=1}^K$.

If $F_t, H_t$ are known, the supervised setting permits a direct calculation of the sample covariance matrices of the noise [Lacey, 1998]:

$$\hat{Q} := Cov(\{x_{k,t+1} - F_t x_{k,t}\}_{k,t}), \qquad \hat{R} := Cov(\{z_{k,t} - H_t x_{k,t}\}_{k,t}). \tag{2}$$

Since Theorem 1 guarantees optimality when $\hat{Q} = Q$, $\hat{R} = R$, and Eq. (2) provides a simple estimator for $Q$ and $R$, Algorithm 1 has become the gold-standard tuning method for KF from supervised data.

**Algorithm 1** (KF noise estimation). Given supervised data $\{(x_{k,t}, z_{k,t})\}$, return $\hat{Q}$ and $\hat{R}$ of Eq. (2).

While Algorithm 1 is indeed trivial to apply in the supervised setting, we show below that when Assumption 1 is violated, it no longer provides optimal predictions. Violation of Assumption 1 can be partially handled by certain variations of the KF, such as Extended KF (EKF) [Sorenson, 1985] and Unscented KF (UKF) [Wan and Van Der Merwe, 2000].

## 3 Optimized Kalman Filter

Estimation of the KF noise parameters $\hat{Q}, \hat{R}$ has been studied extensively in various settings; yet, in our supervised setting it is trivially solved by Algorithm 1. However, once Assumption 1 is violated,

such noise estimation is no longer a proxy to MSE optimization – despite Theorem 1. Instead, in this section we propose to determine $\hat{Q}$ and $\hat{R}$ via explicit MSE optimization. We rely on standard optimization methods for sequential supervised learning; as discussed below, the main challenge is to maintain the Symmetric and Positive Definite (SPD) structure of $\hat{Q}, \hat{R}$ as covariance matrices.

Formally, we consider the KF (Fig. 2) as a prediction model $\hat{x}_{k,t}(\{z_{k,\tau}\}_{\tau=1}^{t}; \hat{Q}, \hat{R})$, which estimates $x_{k,t}$ given the observations $\{z_{k,\tau}\}_{\tau=1}^{t}$ and parameters $\hat{Q}, \hat{R}$. We define the KF optimization problem:

$$\operatorname*{argmin}_{Q', R'} \sum_{k=1}^{K} \sum_{t=1}^{T_k} \operatorname{loss}\left(\hat{x}_{k,t}\left(\{z_{k,\tau}\}_{\tau=1}^{t}; Q', R'\right), x_{k,t}\right), \qquad \text{s.t. } Q' \in S_{++}^{d_x}, R' \in S_{++}^{d_z}, \quad (3)$$

where $S_{++}^{d} \subset \mathbb{R}^{d \times d}$ is the space of Symmetric and Positive Definite matrices (SPD), and $\operatorname{loss}(\cdot)$ is the objective function (e.g., $\operatorname{loss}(\hat{x}, x) = ||\hat{x} - x||^2$ for MSE). Prediction of future states can be expressed using the same Eq. (3), by changing the observed input from $\{z_{k,\tau}\}_{\tau=1}^{t}$ to $\{z_{k,\tau}\}_{\tau=1}^{t-1}$.

A significant challenge in solving Eq. (3) is the SPD constraint. Standard numeric supervised optimization methods (e.g., Adam [Diederik P. Kingma, 2014]) may violate the constraint. While the SPD constraint is often bypassed using diagonal restriction [Li et al., 2019, Formentin and Bittanti, 2014], this may significantly degrade the predictions, as demonstrated in the ablation tests in Appendix B.4. Instead, to maintain the complete expressiveness of $\hat{Q}$ and $\hat{R}$, we use the Cholesky parameterization [Pinheiro and Bates, 1996].

---

**Algorithm 2:** Optimized Kalman Filter (OKF)

---

1 **Input**: training data $\{(x_{k,t}, z_{k,t})\}_{k=1}^{K}$ (Definition 1); batch size $b$; loss function (e.g., MSE); optimization_step function (e.g., Adam)

2 $d_x \leftarrow \operatorname{len}(x_{1,1}), \quad d_z \leftarrow \operatorname{len}(z_{1,1})$
3 Initialize $\theta_Q \in \mathbb{R}^{\frac{1}{2}d_x(d_x+1)}, \theta_R \in \mathbb{R}^{\frac{1}{2}d_z(d_z+1)}$
4 **while** training not finished **do**
  // Get $Q, R$ using Eq. (4)
5 $\quad \hat{Q} \leftarrow L(\theta_Q)L(\theta_Q)^{\top}, \quad \hat{R} \leftarrow L(\theta_R)L(\theta_R)^{\top}$
6 $\quad \mathcal{K} \leftarrow \operatorname{sample}(\{1, ..., K\}, \operatorname{size}=b)$
7 $\quad C \leftarrow 0$
8 $\quad$ **for** $k$ in $\mathcal{K}$ **do**
9 $\quad\quad$ Initialize $\hat{x} \in \mathbb{R}^{d_x}$
10 $\quad\quad$ **for** $t$ in $1 : T_k$ **do**
      // KF steps (Fig. 2)
11 $\quad\quad\quad \hat{x} \leftarrow \operatorname{KF\_predict}(\hat{x}; \hat{Q})$
12 $\quad\quad\quad \hat{x} \leftarrow \operatorname{KF\_update}(\hat{x}, z_{k,t}; \hat{R})$
13 $\quad\quad\quad C \leftarrow C + \operatorname{loss}(\hat{x}, x_{k,t})$
14 $\quad \theta_Q, \theta_R \leftarrow \operatorname{optimization\_step}(C, (\theta_Q, \theta_R))$
15 Return $\hat{Q}, \hat{R}$

---

The parameterization relies on Cholesky decomposition: any SPD matrix $A \in \mathbb{R}^{d \times d}$ can be written as $A = LL^{\top}$, where $L$ is lower-triangular with positive entries along its diagonal. Reversely, for any lower-triangular $L$ with positive diagonal, $LL^{\top}$ is SPD. Thus, to represent an SPD $A \in \mathbb{R}^{d \times d}$, we define $A(L) := LL^{\top}$ and parameterize $L(\theta)$ to be lower-triangular, have positive diagonal, and be differentiable in the parameters $\theta$:

$$(L(\theta))_{ij} := \begin{cases} 0 & \text{if } i < j, \\ e^{\theta_{d(d-1)/2+i}} & \text{if } i = j, \\ \theta_{(i-2)(i-1)/2+j} & \text{if } i > j, \end{cases} \quad (4)$$

where $\theta \in \mathbb{R}^{d(d+1)/2}$.

Both Cholesky parameterization and sequential optimization methods are well known tools. Yet, for KF optimization *from supervised data*, we are not aware of any previous attempts to apply them together, as noise estimation (Algorithm 1) is typically preferred.

We wrap the optimization process in the Optimized KF (**OKF**) in Algorithm 2, which outputs optimized parameters $\hat{Q}, \hat{R}$ for Fig. 2. Note that Algorithm 2 optimizes the state estimation at *current* time $t$. By switching Line 12 and Line 13, the optimization will instead be shifted to state prediction at the *next* time-step (as $\hat{x}$ becomes oblivious to the current observation $z_{k,t}$).

**Lack of theoretical guarantees:** If Assumption 1 cannot be trusted, neither noise estimation (Algorithm 1) nor OKF (Algorithm 2) can guarantee global optimality. Still, OKF pursues the MSE objective using standard optimization tools, which achieved successful results in many non-convex problems [Zhong et al., 2020] over millions of parameters [Devlin et al., 2019]. On the other hand, noise estimation pursues a *conflicting* objective, and guarantees significant sub-optimality in certain scenarios, as analyzed in Section 5.

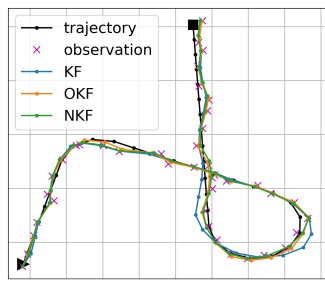

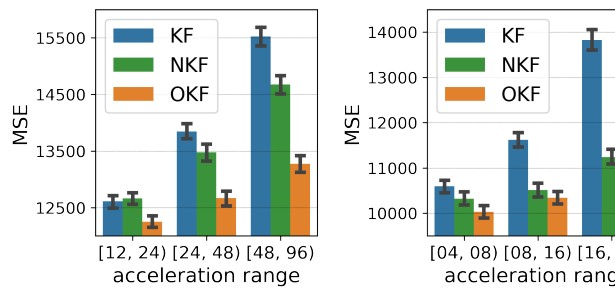

Figure 3: A sample trajectory and the corresponding predictions (projected onto XY plane), in the Free-motion benchmark. The standard KF provides inaccurate predictions in certain turns.

(a) Free-motion benchmark  (b) No-turns benchmark

Figure 4: Test errors and 95% confidence intervals, over targets with different accelerations. The middle acceleration range coincides with the training accelerations (24-48 in (a) and 8-16 in (b)), and the other ranges correspond to out-of-distribution generalization.

## 4 OKF vs. Neural KF: Is the Non-Linearity Helpful?

In this section, we demonstrate that comparing an optimized neural network to a non-optimized baseline may lead to incorrect conclusions: the network may seem superior even if the complicated architecture has no added value. The implied message is not against neural networks, but rather that evaluating them against a non-optimized baseline carries a crucial flaw.

**The Doppler radar problem:** We consider a variant of the classic Doppler radar problem [Barton, 1988, Roy and Mitra, 2016], where various targets trajectories are tracked in a homogeneous 3D space, given regular observations of a Doppler radar. The state $X = (x_x, x_y, x_z, x_{ux}, x_{uy}, x_{uz})^\top \in \mathbb{R}^6$ consists of 3D location and velocity. The goal is to minimize the MSE over the 3 location coordinates. While the true dynamics $F$ are unknown to the KF, a constant-velocity model $\tilde{F}$ can be used:

$$\tilde{F} = \begin{pmatrix} 1 & & 1 & & \\ & 1 & & 1 & \\ & & 1 & & 1 \\ & & & 1 & \\ & & & & 1 \\ & & & & & 1 \end{pmatrix}. \tag{5}$$

An observation $Z \in \mathbb{R}^4$ consists of the location in spherical coordinates (range, azimuth, elevation) and the radial velocity (the Doppler signal), with an additive i.i.d Gaussian noise. After transformation to Cartesian coordinates, the observation model can be written as:

$$H = H(X) = \begin{pmatrix} 1 & & \\ & 1 & \\ & & 1 \\ & & & \frac{x_x}{r} & \frac{x_y}{r} & \frac{x_z}{r} \end{pmatrix}, \tag{6}$$

where $r = \sqrt{x_x^2 + x_y^2 + x_z^2}$. Since $H = H(X)$ relies on the unknown location $(x_x, x_y, x_z)$, we instead substitute $\tilde{H} := H(Z)$ in the KF update step in Fig. 2.

**Neural KF:** The Neural Kalman Filter (NKF) incorporates an LSTM model into the KF framework, as presented in Appendix C and Fig. 16. We originally developed NKF to improve the prediction of the non-linear highly-maneuvering targets in the Doppler problem (e.g., Fig. 3), and made honest efforts to engineer a well-motivated architecture. Regardless, we stress that this section demonstrates a methodological flaw when comparing *any* optimized filtering method to the KF; this methodological argument stands regardless of the technical quality of NKF. In addition, Appendix C presents similar results for other variants of NKF.

**Experiments:** We train NKF and OKF on a dataset of simulated trajectories, representing realistic targets with free motion (as displayed in Fig. 3). As a second benchmark, we also train on a dataset of simplified trajectories, with speed changes but with no turns. The two benchmarks are specified in detail in Appendix B.1, and correspond to Fig. 11d and Fig. 11e. We tune the KF from the same datasets using Algorithm 1. In addition to out-of-sample test trajectories, we also test generalization to out-of-distribution trajectories, generated using different ranges of target accelerations (affecting both speed changes and turns radiuses).

Fig. 4 summarizes the test results. Compared to KF, NKF reduces the errors in both benchmarks, suggesting that the non-linear architecture pays off. However, optimization of the KF (using OKF) reduces the errors even further, and thus reverses the conclusion. That is, the advantage of NKF in this problem comes *exclusively* from optimization, and *not at all* from the expressive architecture.

## 5   OKF vs. KF:

Section 4 presents the methodological contribution of OKF for non-linear filtering, as an optimized baseline for comparison, instead of the standard KF. In this section, we study the advantage of OKF over the KF more generally. We show that OKF consistently outperforms the KF in a variety of scenarios from 3 different domains. This carries considerable practical significance: unlike neural models, shifting from KF to OKF merely requires change of the parameters $\hat{Q}$, $\hat{R}$, hence can be deployed to real-world systems without additional overhead, complexity or latency on inference.

Recall that by Theorem 1, the KF is already optimal unless Assumption 1 is violated. Thus, the violations are discussed in depth, and the effects of certain violations are analyzed theoretically.

### 5.1   Doppler Radar Tracking

Theorem 1 guarantees the optimality of Algorithm 1 (KF). Yet, in Section 4, OKF outperforms the KF. This is made possible by the violation of Assumption 1: while the Doppler problem of Section 4 may not seem complex, the trajectories follow a non-linear motion model (as displayed in Fig. 3).

Imagine that we simplified the problem by only simulating constant-velocity targets, making the true motion model $F$ linear. Would this recover Assumption 1 and make OKF unnecessary? The answer is *no*; the adventurous reader may attempt to list all the remaining violations before reading on.

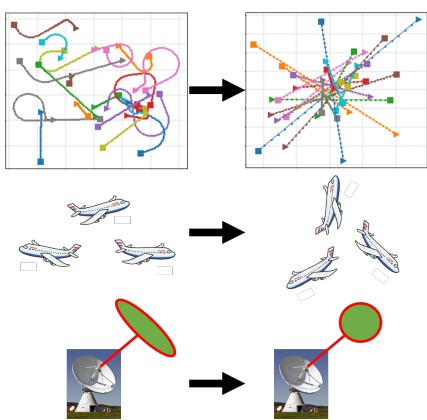

The simulated targets move mostly horizontally, with limited elevation changes. This is not expressed by the KF's initial state distribution $(\hat{x}_0, \hat{P}_0)$. To remedy this, one may simulate motion uniformly in all directions. A third violation comes from the observation noise. While the radar noise is i.i.d in spherical coordinates (as mentioned in Section 4), it is not i.i.d in *Cartesian* coordinates (see discussion in Appendix A.2). To overcome this, one may simulate a radar with (physically-impossible) Cartesian i.i.d noise. This results in the unrealistically-simplified problem visualized in Fig. 5.

Figure 5: The original Doppler problem (left) is simplified to a toy problem (right), with linear motion, isotropic flying directions and physically-impossible radar. After all the simplifications, Assumption 1 still does not hold, thus Algorithm 1 is still sub-optimal and outperformed by OKF.

Despite the simplifications, it turns out that Assumption 1 is still not met, as the observation model in Eq. (6) is still not linear (i.e., $H = H(X)$ is not constant). As shown by Proposition 1, this single violation alone results in a significant deviation of Algorithm 1 from the optimal parameters.

We first define the simplified problem.

**Problem 1** (The toy Doppler problem). The toy Doppler problem is the filtering problem modeled by Eq. (1), with constant-velocity dynamics $F$ (Eq. (5)), Doppler observation $H$ (Eq. (6)), and

$$Q = \mathbf{0} \in \mathbb{R}^{6\times6}, \qquad R = \begin{pmatrix} \sigma_x^2 & & & \\ & \sigma_y^2 & & \\ & & \sigma_z^2 & \\ & & & \sigma_D^2 \end{pmatrix},$$

where $\sigma_x, \sigma_y, \sigma_z, \sigma_D > 0$.

Recall that $H = H(X)$ in Eq. (6) depends on the state $X$, which is unknown to the model. Thus, we assume that $\tilde{H} = H(\tilde{X})$ is used in the KF update step (Fig. 2), with some estimator $\tilde{X} \approx X$ (e.g.,

$\tilde{H} = H(Z)$ in Section 4). Hence, the *effective* noise is $\tilde{R} := Cov(Z - \tilde{H}X) \neq Cov(Z - HX) = R$. Proposition 1 analyzes the difference between $\tilde{R}$ and $R$. To simplify the analysis, we further assume that the error $\tilde{X} - X$ within $\tilde{H}$ (e.g., $Z - X$) is independent of the target velocity.

**Proposition 1.** In the toy Doppler Problem 1 with the estimated observation model $\tilde{H}$, the effective observation noise $\tilde{R} = Cov(Z - \tilde{H}X)$ is:

$$\tilde{R} = \begin{pmatrix} \sigma_x^2 & & & \\ & \sigma_y^2 & & \\ & & \sigma_z^2 & \\ & & & \sigma_D^2 + C \end{pmatrix} = R + \begin{pmatrix} 0 & & & \\ & 0 & & \\ & & 0 & \\ & & & C \end{pmatrix}, \tag{7}$$

where $C = \Omega(\mathbb{E}[||u||^2])$ is the asymptotic lower bound ("big omega") of the expected square velocity $u$. In particular, $C > 0$ and is unbounded as the typical velocity grows.

*Proof sketch (see complete proof in Appendix A.1).* We have $Cov(Z - \tilde{H}X) = Cov(Z - HX + (H - \tilde{H})X) = R + Cov((H - \tilde{H})X)$, where the last equality relies on the independence between the target velocity and the estimation error $\tilde{X} - X$. We then calculate $Cov((H - \tilde{H})X)$. $\square$

Proposition 1 has an intuitive interpretation: when measuring the velocity, Algorithm 1 only considers the inherent Doppler signal noise $\sigma_D$. However, the *effective* noise $\sigma_D + C$ also includes the *transformation error* from Doppler to the Cartesian coordinates, caused by the uncertainty in $H(X)$ itself. Notice that heuristic solutions such as inflation of $R$ would not recover the effective noise $\tilde{R}$, which only differs from $R$ in one specific entry.

Yet, as demonstrated below, OKF captures the effective noise $\tilde{R}$ successfully. Critically, it does so from mere data: OKF does not require the user to specify the model correctly, or to even be aware of the violation of Assumption 1.

**Experiments:** We test KF and OKF on the toy Problem 1 using the same methodology as in Section 4. In accordance with Proposition 1, OKF adapts the Doppler noise parameter: as shown in Fig. 6, it increases $\sigma_D$ in proportion to the location noise by a factor of $\approx 13$. Note that we refer to the proportion instead of absolute values due to scale-invariance in the toy problem, as discussed in Appendix A.1. Following the optimization, **OKF reduces the test MSE by 44%** – from 152 to 84.

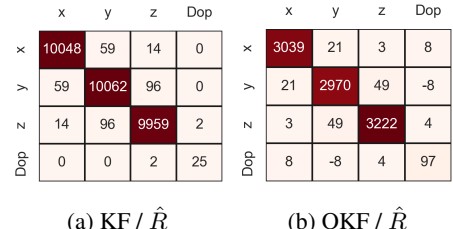

(a) KF / $\hat{R}$      (b) OKF / $\hat{R}$

Figure 6: The parameters $\hat{R}$ learned by KF and OKF in the toy Doppler problem. The rows and columns' entries correspond to location $(x, y, z)$ and radial velocity ($Doppler$). The simulated noise variance is $100^2$ for the positional dimensions and $5^2$ for velocity, and is estimated accurately by the KF. However, OKF increases the noise associated with velocity, in accordance with Proposition 1. The decrease in the positional variance comes from scale-invariance in the toy problem, as discussed in Appendix A.1.

**Extended experiments:** This section and Section 4 test OKF against KF in three specific variants of the Doppler problem. One may wonder if OKF's advantage generalizes to other scenarios, such as:

- Different subsets of violations of Assumption 1;
- Other baseline models than KF, e.g., Optimized Extended-KF;
- Small training datasets;
- Generalization to out-of-distribution test data.

The extended experiments in Appendix B address *all* of the concerns above by examining a wide range of problem variations in the Doppler radar domain. In addition, other domains are experimented below. **In all of these experiments, OKF outperforms Algorithm 1 in terms of MSE**.

Finally, the goal-misalignment of Algorithm 1 is demonstrated directly by two results: even oracle noise estimation fails to optimize the MSE (Appendix B.1); and feeding more data to Algorithm 1 may *degrade* the MSE (Appendix B.2). Fig. 7 presents a sample of the results of Appendix B.2.

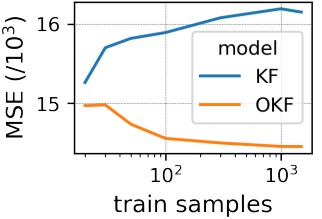

Figure 7: Different train data sizes in the Doppler problem (Fig. 3). Due to objective misalignment, Algorithm 1 deteriorates with more train data.

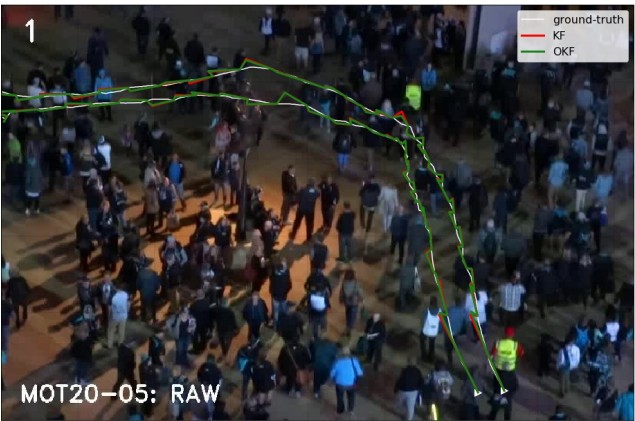

Figure 8: A sample of 2 trajectories in the first frame of MOT20 test video, along with the predictions of KF and OKF.

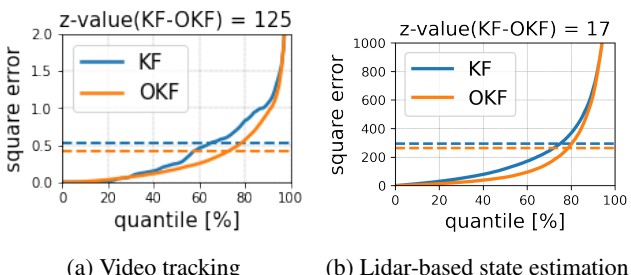

(a) Video tracking      (b) Lidar-based state estimation

Figure 9: Summary of the test errors in the video and lidar problems. The dashed lines correspond to MSE. Both z-values correspond to p-value $< 10^{-6}$. Each z-value is calculated over $N$ test trajectories as follows: $z = \frac{\text{mean}(\{\Delta_i\})}{\text{std}(\{\Delta_i\})}\sqrt{N}$, where $\Delta_i = err_i(KF)^2 - err_i(OKF)^2$ is the square-error difference on trajectory $1 \le i \le N$.

## 5.2 Video Tracking

The MOT20 dataset [Dendorfer et al., 2020] contains videos of real-world targets (mostly pedestrians, as shown in Fig. 8), along with their true location and size in every frame. For our experimental setup, since object detection is out of the scope, we assume that the true locations are known in real-time. The objective is to predict of the target location in the next frame. The state space corresponds to the 2D location, size and velocity, and the observations include only the location and size. The underlying dynamics $F$ of the pedestrians are naturally unknown, and the standard constant-velocity model is used for $\tilde{F}$. This results in the following model:

$$\tilde{F} = \begin{pmatrix} 1 & & 1 & \\ & 1 & & 1 \\ & & 1 & \\ & & & 1 \end{pmatrix}, \quad \tilde{H} = H = \begin{pmatrix} 1 & & 0 & 0 \\ & 1 & 0 & 0 \\ & & 0 & 0 \\ & & 1 & 0 & 0 \end{pmatrix}.$$

Notice that the known observation model $\tilde{H} = H$ is *linear* ($H$ is independent of $X$), hence poses a substantial difference from Section 5.1 in terms of violations of Assumption 1.

The first three videos with 1117 trajectories are used for training, and the last video with 1208 trajectories for testing. As shown in Fig. 8a, OKF reduces the test MSE by 18% with high statistical significance.

## 5.3 Lidar-based State Estimation in Self Driving

Consider the problem of state-estimation in self-driving, based on lidar measurements with respect to known landmarks [Moreira et al., 2020]. The objective is to estimate the current vehicle location. We assume a single landmark (since the landmark matching problem is out of scope). We simulate driving trajectories consisting of multiple segments, with different accelerations and turn radiuses

(see Fig. 15a in the appendix). The state is the vehicle's 2D location and velocity, and $\tilde{F}$ is modeled according to constant-velocity. The observation (both true $H$ and modeled $\tilde{H}$) corresponds to the location, with an additive Gaussian i.i.d noise in polar coordinates. This results in the following model:

$$\tilde{F} = \begin{pmatrix} 1 & 0 & 1 & 0 \\ 0 & 1 & 0 & 1 \\ 0 & 0 & 1 & 0 \\ 0 & 0 & 0 & 1 \end{pmatrix}, \quad \tilde{H} = H = \begin{pmatrix} 1 & 0 & 0 & 0 \\ 0 & 1 & 0 & 0 \end{pmatrix}.$$

We train KF and OKF over 1400 trajectories and test them on 600 trajectories. As shown in Fig. 8b, OKF reduces the test MSE by 10% with high statistical significance.

Notice that the lidar problem differs from Section 5.1 in the linear observation model $H$, and from Section 5.2 in the additive noise. Both properties have a major impact on the problem, as analyzed in Proposition 1 and below, respectively.

**Theoretical analysis:** As mentioned in Section 5.1 and discussed in Appendix A.2, the i.i.d noise in polar coordinates is not i.i.d in Cartesian ones. To isolate the i.i.d violation and study its effect, we define a simplified toy model – with simplified states, no-motion model $F$, isotropic motion noise $Q$ and only radial observation noise. Note that in contrast to Section 5.1, the observation model is already linear.

**Problem 2** (The toy lidar problem). The toy lidar problem is the filtering problem modeled by Eq. (1) with the following parameters:

$$F = H = \begin{pmatrix} 1 & 0 \\ 0 & 1 \end{pmatrix}, \; Q = \begin{pmatrix} q & 0 \\ 0 & q \end{pmatrix}, \; R_{polar} = \begin{pmatrix} r_0 & 0 \\ 0 & 0 \end{pmatrix},$$

for some unknown $q, r_0 > 0$, with observation noise drawn i.i.d from $\mathcal{N}(0, R_{polar})$ in *polar* coordinates. The initial state $X_0$ follows a radial distribution (i.e., with a PDF of the form $f(||x_0||)$).

**Proposition 2.** As the number $N$ of train trajectories in Problem 2 grows, the noise parameter $\hat{R}_N(KF)$ estimated by Algorithm 1 converges almost surely:

$$\hat{R}_N(KF) \xrightarrow{\text{a.s.}} \hat{R}_{est} = \begin{pmatrix} r_0/2 & 0 \\ 0 & r_0/2 \end{pmatrix}.$$

On the other hand, under regularity assumptions, the MSE is minimized by the parameter $\hat{R}_{opt} = \begin{pmatrix} r & 0 \\ 0 & r \end{pmatrix}$, where $r < r_0/2$.

*Proof sketch (see complete proof in Appendix A.2).* For $\hat{R}_{est}$, we calculate $\mathbb{E}[\hat{R}_N(KF)]$ and use the law of large numbers. For the calculation, we transform $R_{polar}$ to Cartesian coordinates using the random direction variable $\theta$, and take the expectation over $\theta \sim U([0, 2\pi))$. The uniform distribution of $\theta$ comes from the radial symmetry of the problem. For $\hat{R}_{opt}$, we calculate and minimize the expected square error directly. $\square$

Intuitively, Proposition 2 shows that the i.i.d violation reduces the *effective* noise. Note that the analysis only holds for the unrealistic toy Problem 2. The empirical setting in this section is less simplistic, and generalizing Proposition 2 is not trivial. Fortunately, OKF optimizes directly from the data, and does not require such theoretical analysis. Fig. 10 shows that indeed, in accordance with the intuition of Proposition 2, OKF learns to reduce the values of $\hat{R}$ in comparison to KF. This results in reduced test errors as specified above.

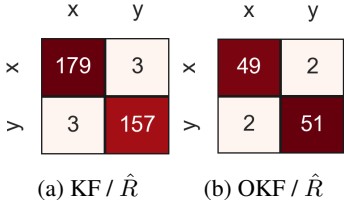

(a) KF / $\hat{R}$      (b) OKF / $\hat{R}$

Figure 10: The parameters $\hat{R}$ learned in the lidar problem. From data alone, OKF learns to decrease the noise parameters, consistently with Proposition 2.

# 6 Related Work

**Noise estimation** of the KF parameters from observations alone has been studied for decades, as supervised data (Definition 1) is often unavailable. Various methods were studied, based on autocorrelation [Mehra, 1970, Carew and Belanger, 1973], EM [Shumway and Stoffer, 2005] and others [Odelson et al., 2006, Feng et al., 2014, Park et al., 2019]. When supervised data *is* available, noise estimation reduces to Eq. (2) and is considered a solved problem [Odelson et al., 2006]. We show that while noise estimation is indeed easy from supervised data, it may be the wrong objective to pursue.

**Optimization:** We apply gradient-based optimization to the KF with respect to its errors. In absence of supervised data, gradient-based optimization was suggested for other losses, such as smoothness [Barratt and Boyd, 2020]. In the supervised setting, noise estimation is typically preferred [Odelson et al., 2006], although optimization without gradients was suggested in Abbeel et al. [2005]. In practice, "optimization" of KF is sometimes handled by trial and error [Jamil et al., 2020] or grid search [Formentin and Bittanti, 2014, Coskun et al., 2017]. In other cases, $Q$ and $R$ are restricted to be diagonal [Li et al., 2019, Formentin and Bittanti, 2014]. However, such heuristics may not suffice when the optimal parameters take a non-trivial form.

**Neural Networks (NNs) in filtering:** The NKF in Section 4 relies on a recurrent NN. NNs are widely used in non-linear filtering, e.g., for online prediction [Gao et al., 2019, Iter et al., 2016, Coskun et al., 2017, fa Dai et al., 2020, Belogolovsky et al., 2022], near-online prediction [Kim et al., 2018], and offline prediction [Liu et al., 2019b]. Learning visual features for tracking via a NN was suggested by Wojke et al. [2017]. NNs were also considered for related problems such as data association [Liu et al., 2019a], model switching [Deng et al., 2020], and sensors fusion [Sengupta et al., 2019].

In addition, all the 10 studies cited in Section 1 used a NN model for non-linear filtering, with either KF or EKF as a baseline for comparison. As discussed above, none has optimized the baseline model to a similar extent as the NN. As demonstrated in Section 4, such methodology could lead to unjustified conclusions.

# 7 Summary

We observed that violation of the KF assumptions is common, and is potentially difficult to notice or model. Under such violation, we analyzed (theoretically and empirically) that the standard noise estimation of the KF parameters conflicts with MSE optimization. An immediate consequence is that the KF is often used sub-optimally. A second consequence is that in many works in the literature, where a neural network is compared to the KF, the experiments become inconclusive: they cannot decide whether the network succeeded due to superior architecture, or merely because its parameters were optimized. We presented the Optimized KF (OKF), and demonstrated that it can solve both issues (Section 5 and Section 4, respectively).

From a practical point of view, the OKF is available on PyPI and is easily applicable to new problems. Since its architecture is identical to the KF, and only the parameters are changed, the learned model causes neither inference-time delays nor deployment overhead. All these properties make the OKF a powerful practical tool for both linear and non-linear filtering problems.

**Acknowledgements**

The authors thank Tal Malinovich, Ophir Nabati, Zahar Chikishev, Mark Kozdoba, Eli Meirom, Elad Sharony, Itai Shufaro and Shirli Di-Castro for their helpful advice. This work was partially funded by the European Union's Horizon Europe Programme, under grant number 101070568.

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

# Contents

# A Theoretical Analysis

## A.1 Non-linear Observation

In this section, we discuss the relation between the theoretical analysis of Proposition 1 and the empirical results shown in Fig. 6. Then, we provide the proof of Proposition 1.

**Fig. 6 vs. Proposition 1:** Fig. 6 displays the noise parameters $\hat{R}$ learned by OKF in the toy problem. In accordance with Proposition 1, the noise $\sigma_D$ associated with Doppler is increased compared to the true measurement noise $R$. In fact, not only $\sigma_D$ is increased, but also the positional variances are decreased, which is not explained by Proposition 1. This phenomenon origins in the absence of dynamics noise in this toy problem ($Q \equiv 0$), which leads to scale-invariance w.r.t. the absolute values of $\hat{R}$. That is, if we multiply the whole matrix $\hat{R}$ by a constant factor, the filtering errors are unaffected. Specifically, if we multiply $\hat{R}$ of Fig. 6b by a factor of $\approx 3$, the positional variances become aligned with those of Fig. 6a, and $\sigma_D$ is increased by a factor of $\approx 13$ – in accordance with Proposition 1. We repeated the tests with this modified $\hat{R}$, and indeed, the results were indistinguishable from the original OKF.

*Proof of Proposition 1.* Recall that in this problem, the KF applies the update step using an estimated observation model $\tilde{H} = H(\tilde{X})$:

$$\tilde{H} = \begin{pmatrix} 1 & & & \\ & 1 & & \\ & & 1 & \\ & & & \tilde{x}_x/\tilde{r} \quad \tilde{x}_y/\tilde{r} \quad \tilde{x}_z/\tilde{r} \end{pmatrix}.$$

Denoting the normalized estimation error $dx' = \frac{\tilde{x}}{\tilde{r}} - \frac{x}{r}$, we can rewrite $\tilde{H}$ as

$$\tilde{H} = H + \begin{pmatrix} 0 & & & \\ & 0 & & \\ & & 0 & \\ & & & dx'_x \quad dx'_y \quad dx'_z \end{pmatrix}.$$

By shifting the observation model in Eq. (1) from $H$ to $\tilde{H}$, and denoting the noise by $\nu = (\nu_x, \nu_y, \nu_z, \nu_D)^\top$, we receive

$$Z = HX + \nu = \tilde{H}X + \begin{pmatrix} \nu_x \\ \nu_y \\ \nu_z \\ \nu_D - dx'_x u_x - dx'_y u_y - dx'_z u_z \end{pmatrix} = \tilde{H}X + \begin{pmatrix} \nu_x \\ \nu_y \\ \nu_z \\ \nu_D - dx' \cdot u \end{pmatrix},$$

where $u$ denotes the current target velocity. We see that the effective observation noise is $\tilde{\nu} = Z - \tilde{H}X = (\nu_x, \nu_y, \nu_z, \nu_D - dx' \cdot u)^\top$.

To show that all the off-diagonal entries of $\tilde{R} = Cov(\tilde{\nu})$ vanish, recall that the estimation error $dx'$ is assumed to be independent of the velocity $u$. According to Eq. (1), $\nu$ is also independent of $u$. Hence, $Cov(dx'_x \cdot u_x, \ \nu_x) = E(dx'_x \cdot u_x \cdot \nu_x) = E(dx'_x \nu_x)E(u_x)$ which vanishes by symmetry ($E(u_x) = 0$). The same result holds for coordinates $y, z$. Thus, $\tilde{R}$ is diagonal. Finally, by denoting $C = Var(dx' \cdot u) > 0$ we have $Cov(\tilde{\nu}) = \tilde{R}$ as required.

Relying again on symmetry $E(u), E(dx') = 0$, we can further calculate $C = Var(dx' \cdot u) = E(||dx'||^2)E(||u||^2) = \Omega(E(||u||^2))$, where $\Omega$ ("big-omega") corresponds to an asymptotic lower bound. $\qquad\square$

## A.2 Non-i.i.d Noise

The assumption of i.i.d noise in Assumption 1 is violated in many practical scenarios. Certain models with non-i.i.d noise can be solved analytically, if modeled correctly. For example, if the noise is auto-regressive with a known order $p$, an adjusted KF model may consider the last $p$ values of the noise itself as part of the system state [Geist and Pietquin, 2011]. However, the actual noise model is often unknown or infeasible to solve analytically.

Furthermore, the violation of the i.i.d assumption may even go unnoticed. We discuss a potential example in Section 5, where the noise is i.i.d in *spherical* coordinates – but is not so after the transformation to *Cartesian* coordinates. To see that, consider a radar with noiseless angular estimation (i.e.,

only radial noise), and a low target ($x_z \approx 0$). Clearly, most of the noise concentrates on the XY plane – both in the current time-step and in the following ones (until the target moves away from the plane). Hence, the noise is statistically-dependent over time-steps.

We may formalize this intuition for the toy Problem 2. Denote the system state at time $t$ by $X_t = ((X_t)_1, (X_t)_2)^\top$, and denote $\tan \theta_t = \frac{(X_t)_2}{(X_t)_1}$. By transforming $R_{polar}$ of Problem 2 to Cartesian coordinates, the observation noise is drawn from the distribution $\nu_t \sim \mathcal{N}(0, R(\theta_t))$, where

$$R(\theta) = \begin{pmatrix} r_0 \cos^2(\theta) & r_0 \cos(\theta) \sin(\theta) \\ r_0 \cos(\theta) \sin(\theta) & r_0 \sin^2(\theta) \end{pmatrix}. \tag{8}$$

Since consecutive time steps are likely to have similar values of $\theta_t$, the noise $\nu_t$ is no longer independent across time steps.

The effect of this violation of the i.i.d assumption is analyzed in Proposition 2, whose proof is provided below.

*Proof of Proposition 2.*

**Noise estimation:** First, notice that the whole setting of Problem 2 is invariant to the target direction $\theta$: the initial state distribution is radial, and the motion noise $Q$ is isotropic. Hence, for any target at any time-step, $\theta_t \sim [0, 2\pi)$ is uniformly distributed. By direct calculation,

$$E_\theta \left[ \hat{R}_N(KF)_{11} \right] = E_\theta \left[ r_0 \cos^2 \theta \right] = \int_0^{2\pi} \frac{r_0}{2\pi} \cos^2 \theta d\theta = \frac{r_0}{2}$$

$$E_\theta \left[ \hat{R}_N(KF)_{22} \right] = E_\theta \left[ r_0 \sin^2 \theta \right] = \int_0^{2\pi} \frac{r_0}{2\pi} \sin^2 \theta d\theta = \frac{r_0}{2}$$

$$E_\theta \left[ \hat{R}_N(KF)_{12} \right] = E_\theta \left[ \hat{R}_N(KF)_{21} \right] = E_\theta \left[ r_0 \cos \theta \sin \theta \right] = 0.$$

Since the targets in the data are i.i.d, the noise estimation of Algorithm 1 converges almost surely according to the law of large numbers, as required:

$$\hat{R}_N(KF) \xrightarrow{\text{a.s.}} \hat{R}_{est} = \begin{pmatrix} r_0/2 & 0 \\ 0 & r_0/2 \end{pmatrix}.$$

**Optimization:** We use again the radial symmetry and invariance to rotations in the problem: w.l.o.g, we assume that the optimal noise covariance parameter is diagonal, i.e., $\hat{R}_{opt}(r) = \begin{pmatrix} r & 0 \\ 0 & r \end{pmatrix}$ for some $r > 0$. Our goal is to find $r$, and in particular to compare it to $r_0/2$.

At a certain time $t$, where the system state is $X_t$, denote $E[X_t] = x_0 = (x_1, x_2)^\top$ and $Cov(X_t) = P_0 = \begin{pmatrix} p & 0 \\ 0 & p \end{pmatrix}$ (where $p > 0$). Denote the observation received at time $t$ by $z = (x_1 + dx_1, x_2 + dx_2)^\top$. We are interested in the point-estimate $\hat{x}$ of the KF following the update step (Fig. 2). By substituting $x_0, P_0$, the observation $z$ and the noise parameter $\hat{R}_{opt}(r)$ in the update step, we have

$$\hat{x} = x_0 + P_0 H^\top (H P_0 H^\top + \hat{R}_{opt}(r))^{-1}(z - Hx_0) = x_0 + P_0(P_0 + \hat{R}_{opt}(r))^{-1}(z - x_0)$$

$$= x_0 + \begin{pmatrix} \frac{p}{p+r} & 0 \\ 0 & \frac{p}{p+r} \end{pmatrix} \begin{pmatrix} dx_1 \\ dx_2 \end{pmatrix} = \begin{pmatrix} x_1 + \frac{p}{p+r} dx_1 \\ x_2 + \frac{p}{p+r} dx_2 \end{pmatrix}.$$

On the other hand, the *true* observation noise covariance at time $t$ is $R(\theta_t)$ of Eq. (8) (for the random variable $\theta_t$). If we add the assumption that the state $X_t$ is normally distributed ($X_t \sim \mathcal{N}(x_0, P_0)$), and use the true noise covariance $R(\theta_t)$, then the update step of Fig. 2 gives us the true posterior expected state:

$$x_{true} = x_0 + P_0(P_0 + R(\theta_t))^{-1}(z - x_0)$$

$$= x_0 + \begin{pmatrix} \frac{r_0 \sin^2 \theta + p}{p+r_0} & -\frac{r_0 \cos \theta \sin \theta}{p+r_0} \\ -\frac{r_0 \cos \theta \sin \theta}{p+r_0} & \frac{r_0 \cos^2 \theta + p}{p+r_0} \end{pmatrix} \begin{pmatrix} dx_1 \\ dx_2 \end{pmatrix}$$

$$= \begin{pmatrix} x_1 + \frac{(r_0 \sin^2 \theta + p)dx_1 - (r_0 \cos \theta \sin \theta)dx_2}{p+r_0} \\ x_2 + \frac{(r_0 \cos^2 \theta + p)dx_2 - (r_0 \cos \theta \sin \theta)dx_1}{p+r_0} \end{pmatrix}.$$

We can use the standard MSE decomposition for the point-estimate $x$, into the bias term of $x$ and the variance term of the state distribution: $MSE = MSE_{var}(P_{true}) + MSE_{bias}(x, x_{true})$. Notice that $MSE_{var}(P_{true})$ is independent of our estimator, as it corresponds to the inherent uncertainty $P_{true}$ (defined by applying to $P_0$ the update step with the true covariance $R(\theta_t)$). Thus, our objective is to minimize $MSE_{bias}(\hat{x}, x_{true}) = E[||\hat{x} - x_{true}||^2]$.

For the calculation below, we denote $a(r) := p/(p + r)$ and use the identity $\sin 2\theta = 2\cos\theta\sin\theta$. In addition, from radial symmetry of $dx = z - x_0$ we have $E[dx_1^2] = E[dx_2^2]$ and $E[dx_i] = 0$, thus we can denote $v := Var(dx_i) = E[dx_i^2]$.

$$MSE_{bias}(\hat{x}(a), x_{true}) = E||\hat{x}(a) - x_{true}||^2$$

$$= E\left[\left((a - \frac{r_0\sin^2\theta + p}{p + r_0})dx_1 + \frac{r_0\sin(2\theta)/2}{p + r_0}dx_2\right)^2\right.$$

$$\left. + \left((a - \frac{r_0\cos^2\theta + p}{p + r_0})dx_2 + \frac{r_0\sin(2\theta)/2}{p + r_0}dx_1\right)^2\right]$$

$$= E\left[dx_1^2\left(a^2 - 2a\frac{r_0\sin^2\theta + p}{p + r_0} + C_1\right) + \frac{r_0^2\sin^2(2\theta)/4}{(p + r_0)^2}dx_2^2 + A_1 dx_1 dx_2\right.$$

$$\left. + dx_2^2\left(a^2 - 2a\frac{r_0\cos^2\theta + p}{p + r_0} + C_2\right) + \frac{r_0^2\sin^2(2\theta)/4}{(p + r_0)^2}dx_1^2 + A_2 dx_1 dx_2\right]$$

$$= 2va^2 - 2va\frac{r_0 + 2p}{p + r_0} + v(C_1 + C_2) + v\frac{r_0^2\sin^2(2\theta)/2}{(p + r_0)^2},$$

where $C_{1,2}$ are independent of $a$, and $A_{1,2}$ are multiplied by $E[dx_1 dx_2] = 0$ and vanish. To minimize we calculate

$$0 = \frac{\partial MSE_{bias}(\hat{x}(a), x_{true})}{\partial a} = 4v \cdot a - 2v\frac{2p + r_0}{p + r_0},$$

which gives us

$$a = \frac{p + r_0/2}{p + r_0}.$$

Notice that $MSE_{bias}$ clearly diverges as $|a| \to \infty$, hence the only critical point necessarily corresponds to a minimum of the $MSE$. Hence, the optimal $MSE$ is given when substituting the following $r$ in $\hat{R}_{opt}$:

$$r = p/a - p = \frac{p^2 + pr_0 - (p^2 + pr_0/2)}{p + r_0/2} = \frac{pr_0}{2p + r_0}.$$

Finally, recall that $(\hat{R}_{est})_{ii} = r_0/2$ and compare to $r$ directly:

$$(\hat{R}_{est})_{ii} - (\hat{R}_{opt})_{ii} = r_0/2 - r = \frac{r_0^2/2}{2p + r_0} > 0.$$

$\square$

# B  OKF: Extended Experiments

## B.1  Additional Scenarios and Baselines: A Case Study

In this section, we extend the experiments of Section 5.1 with a detailed case study. The case study considers 5 types of tracking scenarios (*benchmarks*) and 4 variants of the KF (*baselines*) – 20 experiments in total. In each experiment, we compare the test MSE of OKF against the standard KF. The experiments in Section 4 and Section 5.1 are 3 particular cases. For each benchmark, we simulate 1500 targets for training and 1000 targets for testing.

**Benchmarks (scenarios):** Section 5 discusses the sensitivity of Algorithm 1 to violations of Assumption 1. In this case study, we consider 5 benchmarks with different subsets of violations of

Table 1: Benchmarks and the properties that define them. "V" means that the benchmark satisfies the property.

| Benchmark | anisotropic | polar | uncentered | acceleration | turns |
|-----------|-------------|-------|------------|--------------|-------|
| Toy | O | O | O | O | O |
| Close | V | V | O | O | O |
| Const_v | V | V | V | O | O |
| Const_a | V | V | V | V | O |
| Free | V | V | V | V | V |

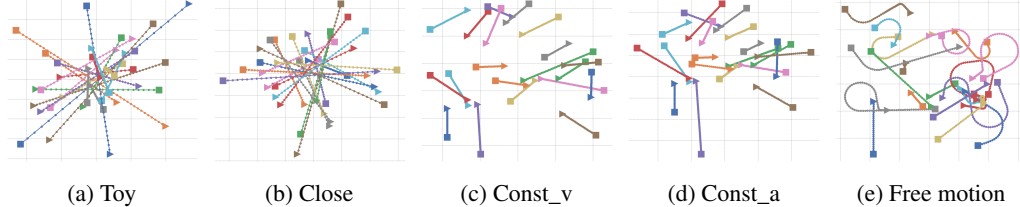

| (a) Toy | (b) Close | (c) Const_v | (d) Const_a | (e) Free motion |
|---------|-----------|-------------|-------------|-----------------|

Figure 11: Samples of targets trajectories in the various benchmarks, projected onto the XY plane.

Assumption 1. The *Free Motion* benchmark is intended to represent a realistic Doppler radar problem, with targets and observations simulated as in Section 4: each target trajectory consists of multiple segments of different turns and accelerations. On the other extreme, the *Toy* benchmark (Problem 1) introduces multiple simplifications (as visualized in Fig. 5). In the Toy benchmark, the only violation of Assumption 1 is the non-linear observation $H$, as discussed in Section 5.1. Note that Section 5.2 and Section 5.3 experiment with settings of a linear observation model.

We design 5 benchmarks within the spectrum of complexity between Toy and Free Motion. Each benchmark is defined as a subset of the following properties, as specified in Table 1 and visualized in Fig. 11:

- *anisotropic*: horizontal motion is more likely than vertical (otherwise direction is distributed uniformly).
- *polar*: radar noise is generated i.i.d in spherical coordinates (otherwise noise is Cartesian i.i.d).
- *uncentered*: targets are dispersed in different locations far from the radar (otherwise they are concentrated in the center).
- *acceleration*: speed change is allowed (through intervals of constant acceleration).
- *turns*: non-straight motion is allowed.

**Baselines (KF variants):** All the experiments above compare OKF to the standard KF baseline. In practice, other variants of the KF are often in use. Here we define 4 such variants as different baselines to the experiments. In each experiment, we compare the baseline tuned by Algorithm 1 to its Optimized version trained by Algorithm 2 (denoted with the prefix "O" in its name). For Algorithm 2, we use the Adam optimizer with a single training epoch over the 1500 training trajectories, 10 trajectories per training batch, and learning rate of 0.01. The optimization was run for all baselines in parallel and required a few minutes per benchmark, on eight i9-10900X CPU cores in a single Ubuntu machine.

The different baselines are designed as follows. *EKF* baselines use the non-linear Extended KF model [Sorenson, 1985]. The EKF replaces the approximation $H \approx H(z)$ of Section 4 with $H \approx \nabla_x h(\hat{x})$, where $h(x) = H(x) \cdot x$ and $\tilde{x}$ is the current state estimate. *Polar* baselines (denoted with "p") represent the observation noise $R$ with spherical coordinates, in which the polar radar noise is i.i.d.

**Results:** Table 2 summarizes the test errors (MSE) in all the experiments. In each cell, the left column corresponds to the baseline Algorithm 1, and the right to Algorithm 2. In the model names,

Table 2: Test MSE results of Algorithm 1 and Algorithm 2 over 5 benchmarks (scenarios) and 4 baselines (variants of KF). For KFp we also consider an "oracle" baseline with perfect knowledge of the noise.

| Benchmark | KF | OKF | KFp | KFp (oracle) | OKFp | EKF | OEKF | EKFp | OEKFp |
|-----------|-----|------|------|------|------|------|------|------|-------|
| Toy | 151.7 | 84.2 | 269.6 | – | 116.4 | 92.8 | **79.4** | 123.0 | 109.1 |
| Close | 25.0 | 24.8 | 22.6 | 22.5 | **22.5** | 26.4 | 26.1 | 24.5 | 24.1 |
| Const_v | 90.2 | 90.0 | 102.3 | 102.3 | **89.2** | 102.5 | 99.7 | 112.7 | 102.1 |
| Const_a | 107.5 | 101.6 | 118.4 | 118.3 | **100.3** | 110.0 | 107.0 | 126.0 | 108.7 |
| Free | 125.9 | 118.8 | 145.6 | 139.3 | **117.9** | 135.8 | 121.9 | 149.3 | 120.0 |

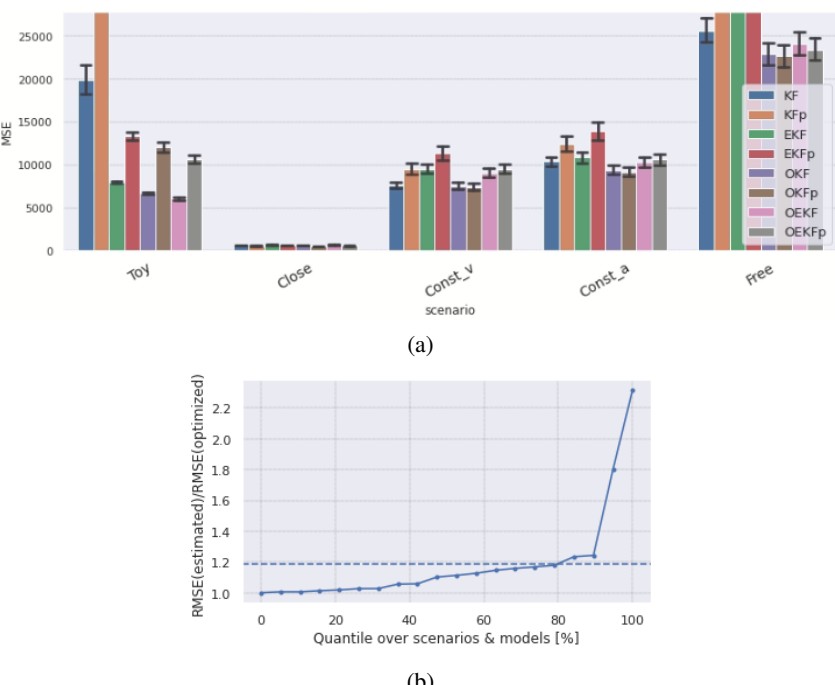

(a)

(b)

Figure 12: Summary of the test MSE of Algorithm 1 and Algorithm 2 in different benchmarks (scenarios) and baselines. This is a different presentation of the results of Table 2. (a) also includes 95% confidence intervals. (b) shows, for each of the 20 experiments (5 benchmarks × 4 baselines), the MSE ratio between Algorithm 1 and Algorithm 2. We see that Algorithm 2 wins in *all* the experiments (ratio is always larger than 1) – in some cases by large margins. The dashed line represents the average MSE ratio over over all the experiments, showing an average advantage of 20% to OKF.

"O" stands for optimized, "E" for EKF and "p" for polar (or spherical). The same results are also shown with confidence intervals in Fig. 12. Below we discuss the main findings.

**Choosing the KF configuration is not trivial:** Consider the non-optimized KF baselines (left column in every cell in Table 2). In each benchmark, the results are sensitive to the baseline, i.e., to the choice of KF configuration – $R$'s coordinates and whether to use EKF. For example, in the Toy benchmark, EKF is the best design, since the observation model $H$ is non-linear. In other benchmarks, however, the winning baselines may come as a surprise:

1. Under non-isotropic motion direction (all benchmarks except Toy), EKF is worse than KF despite the non-linearity. It is possible that the horizontal prior reduces the stochasticity of $H$, making the derivative-based approximation unstable.

2. Even when the observation noise is spherical i.i.d, spherical representation of $R$ is not beneficial when targets are scattered far from the radar (last 3 benchmarks). It is possible that with distant targets, Cartesian coordinates have a more important role in expressing the horizontal prior of the motion.

Since the best KF variant per benchmark seems hard to predict in advance, a practical system cannot rely on choosing the KF variant optimally – and should rather be robust to this choice.

**OKF is more accurate *and* more baseline-robust:** For *every* benchmark and *every* baseline (20 experiments in total), OKF (right column) outperformed noise estimation (left column). In addition, the variance between the baselines reduces under optimization, i.e., OKF makes the KF more robust to the selected configuration.

**OKF outperforms an *oracle* baseline:** We designed an "oracle" KF baseline – with perfect knowledge of the observation noise covariance $R$ in spherical coordinates. We used it for all benchmarks except for Toy (in which the radar noise is not generated in spherical coordinates). Note that in the constant-speed benchmarks (Close and Const_v), $Q = 0$ and is estimated quite accurately; hence, in these benchmarks the oracle has a practically perfect knowledge of both noise covariances. Nevertheless, the oracle yields very similar results to Algorithm 1. This indicates that **the benefit of OKF is not in a better estimation accuracy of $Q$ and $R$, but rather in optimizing the desired objective**.

### B.2  Sensitivity to Train Dataset Size

Each benchmark in the case-study of Appendix B.1 has 1500 targets in its train data. One may argue that numeric optimization may be more sensitive to smaller datasets than noise estimation; and even more so, when taking into account that the optimization procedure "wastes" a portion of the train data as a validation set.

In this section we test this concern empirically, by repeating some of the experiments of Appendix B.1 with smaller subsets of the train datasets – beginning from as few as 20 training trajectories. Fig. 13 shows that the advantage of OKF over KF holds consistently for all sizes of train datasets, although it is indeed increases with the size. Interestingly, in the Free Motion benchmark, **the test errors of KF and KFp *increase* with the amount of train data!**

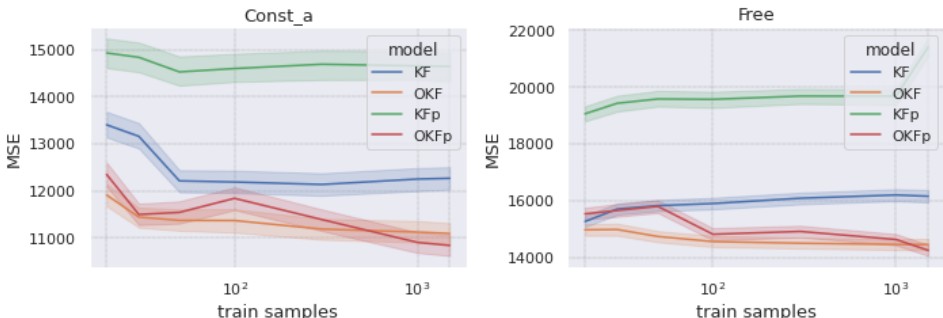

Figure 13: The advantage of OKF over KF holds consistently for all sizes of train datasets – including as small datasets as 20 trajectories. The shadowed areas correspond to 95% confidence intervals.

### B.3  Generalization: Sensitivity to Distributional Shifts

In Appendix B.1, we demonstrate the robustness of OKF in different tracking scenarios: in every benchmark, OKF outperformed the standard KF over **out-of-sample test data**. This means that OKF did not overfit the noise in the training data. What about **out-of-distribution test data**? OKF learns patterns from the specific distribution of the train data – how well will it generalize to different distributions?

Section 4 already addresses this question to some extent, as OKF outperformes both KF and NKF over out-of-distribution target accelerations (affecting both speed changes and turns radius). In terms of Eq. (1), the modified acceleration corresponds to different magnitudes of motion noise $Q$; that is, we change the noise *after* OKF optimized the noise parameters. Yet, OKF adapted to the change without further optimization, with better accuracy than the standard KF. Thus, the results of Section 4 already provide a significant evidence for the robustness of OKF to certain distributional shifts.

In this section, we present a yet stronger evidence for the robustness of OKF – not over a parametric distributional shift, but over **entirely different benchmarks**. Specifically, we consider the 5

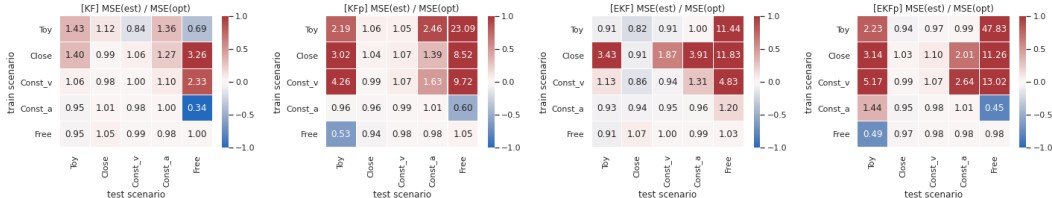

(a) $MSE\_ratio = MSE(KF)/MSE(OKF)$ for every KF-baseline (KF,KFp,EKF,EKFp defined in Appendix B.1), and for every pair of train-scenario and test-scenario. The colormap scale is logarithmic ($\propto log(MSE\_ratio)$), where red values represent advantage to OKF ($MSE\_ratio > 1$).

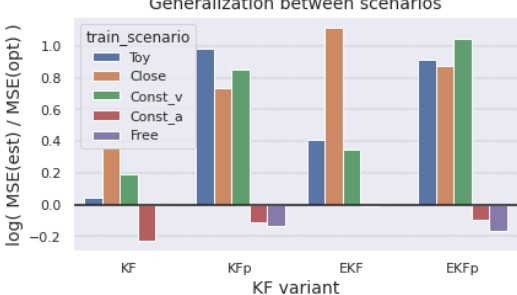

(b) For every train-scenario, $MSE\_ratio$ is averaged over all the test-scenarios and is shown in a logarithmic scale. Positive values indicate advantage to OKF.

Figure 14: Generalization tests: OKF vs. KF under distributional shifts between scenarios.

benchmarks (or scenarios) of Appendix B.1. For every pair (train-scenario, test-scenario), we train both KF and OKF on data of the train-scenario, then test them on data of the test-scenario. For every such pair of scenarios, we measure the generalization advantage of OKF over KF through $MSE\_ratio = MSE(KF)/MSE(OKF)$ (where $MSE\_ratio > 1$ indicates advantage to OKF). To measure the total generalization advantage of a model trained on a certain scenario, we calculate the geometric mean of $MSE\_ratio$ over all the test-scenarios (or equivalently, the standard mean over the logs of the ratios). The logarithmic scale guarantees a symmetric view of this metric of ratio between two scores.

This test is quite noisy, since a model optimized for a certain scenario may legitimately be inferior in other scenarios. Yet, considering all the results together in Fig. 14, it is evident that OKF provides more robust models: it generalizes better in most cases, sometimes by a large margin; and loses only in a few cases, always by a small margin.

## B.4   Ablation Test: Diagonal Optimization

The main challenge in Section 3 and Algorithm 2 is to apply standard numeric optimization while keeping the symmetric and positive-definite constraints (SPD) of the parameters $Q, R$. To that end, we apply the Cholesky parameterization to $Q$ and $R$. In this section, we study the importance of this parameterization via an ablation test.

For the ablation, we define a naive version of OKF with a diagonal parameterization of $Q$ and $R$. Such parameterization is common in the literature: "since both the covariance matrices must be constrained to be positive semi-definite, $Q$ and $R$ are often parameterized as diagonal matrices" [Formentin and Bittanti, 2014]. We denote the diagonal variant by *DKF*, and test it on all 5 Doppler benchmarks of Appendix B.1.

Table 3 displays the results. In the first 3 benchmarks, where the target motion is linear, DKF is indistinguishable from OKF. However, in the 2 non-linear benchmarks (which are also the same benchmarks used in Section 4), DKF is inferior to OKF, though still outperforms the standard KF.

Table 3: Ablation test: Diagonal optimized KF, tested on the 5 benchmarks of Appendix B.1.

| Benchmark | KF | DKF | OKF |
|-----------|------|-------|-------|
| Toy | 151.7 | 84.2 | 84.2 |
| Close | 25.0 | 24.8 | 24.8 |
| Const_v | 90.2 | 90.1 | 90.0 |
| Const_a | 107.5 | 106.1 | 101.6 |
| Free | 125.9 | 121.1 | 118.8 |

## B.5 Video Tracking: Dataset License

The MOT20 video dataset [Dendorfer et al., 2020] is available under *Creative Commons Attribution-NonCommercial-ShareAlike 3.0* License. In Section 5.2, we used the videos MOT20-01, MOT20-02 and MOT20-03 for training, and MOT20-05 for testing.

## B.6 Lidar-based State Estimation: Visualization

Fig. 15 visualizes a sample of the simulated trajectories and the model predictions in the lidar-based state estimation of Section 5.3.

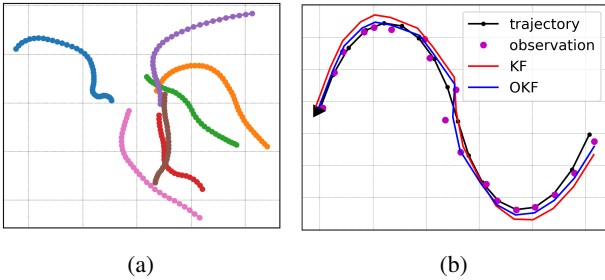

|     |     |
|-----|-----|
| (a) | (b) |

Figure 15: (a) A sample of simulated self-driving trajectories. (b) Segments of turns within a sample trajectory, and the corresponding lidar-based estimations.

## C  Neural KF: Extended Discussion and Experiments

**Preliminaries – RNN and LSTM:** *Recurrent neural networks* (RNN) [Rumelhart et al., 1986] are neural networks that are intended to be iteratively fed with sequential data samples, and that pass information (the *hidden state*) over iterations. Every iteration, the hidden state is fed to the next copy of the network as part of its input, along with the new data sample. *Long Short Term Memory* (LSTM) [Hochreiter and Schmidhuber, 1997] is an architecture of RNN that is particularly popular due to the linear flow of the hidden state over iterations, which allows to capture memory for relatively long term. The parameters of a RNN are usually optimized in a supervised manner with respect to a training dataset of input-output pairs.

**Neural Kalman Filter:** We introduce the Neural Kalman Filter (NKF), which incorporates an LSTM model into the KF framework. The framework provides a probabilistic representation (rather than point estimate) and a separation between the prediction and update steps. The LSTM is an architecture of recurrent neural networks, and is a key component in many SOTA algorithms for non-linear sequential prediction [Neu et al., 2021]. We use it for the non-linear motion prediction.

As shown in Fig. 16, NKF uses separate LSTM networks for prediction and update steps. In the prediction step, the target *acceleration* is predicted on top of the linear motion model, instead of predicting the state directly. This regularized formulation is intended to express our domain knowledge about the kinematic motion of physical targets.

**Extended experiments:** We extend the experiments of Section 4 with additional versions of NKF:

- Predicted-acceleration KF (**aKF**): a variant of NKF that predicts the acceleration but not the covariances $Q$ and $R$.

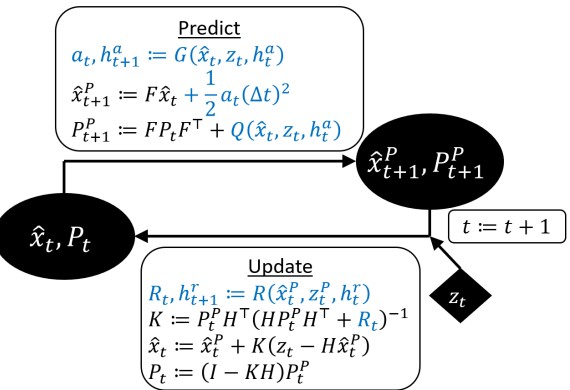

Figure 16: The Neural Kalman Filter (NKF). Differences from Fig. 2 are highlighted. $\Delta t$ is constant; $G, Q$ are the outputs of an LSTM network with hidden state $h_a$; and $R$ is the output of an LSTM with hidden state $h_r$.

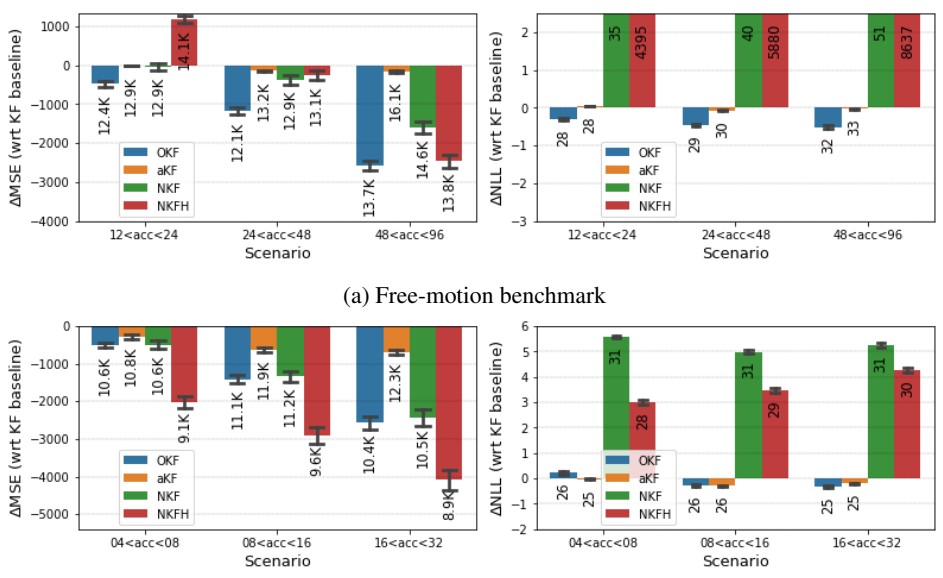

(a) Free-motion benchmark

(b) Const_a benchmark (no turns)

Figure 17: The *relative* MSE and NLL results of various models in comparison to the standard KF model. The textual labels specify the *absolute* MSE and NLL. Note that certain bars of NLL are of entirely different scale and thus are cropped in the figure (their values can be seen in the labels). In each benchmark, the models were trained with relation to MSE loss, on train data of the middle acceleration-range: the two other acceleration ranges in each benchmark correspond to generalization over distributional shifts.

- Neural KF (**NKF**): the model used in Section 4 and illustrated in Fig. 16.
- Neural KF with H-prediction (**NKFH**): a variant of NKF that also predicts the observation model $H$ in every step.

In addition, while we still train with MSE loss, we add the test metric of Negative-Log-Likelihood (NLL) – of the true state w.r.t the estimated distribution. Note that the NLL has an important role in the multi-target matching problem (which is out of the scope of this work).

For each benchmark and each model, we train the model on train data with a certain range of targets acceleration (note that acceleration affects both speed changes and turns sharpness), and tested it on targets with different acceleration ranges, some of them account for distributional shifts. For each model we train two variants – one with Cartesian representation of the observation noise $R$, and one with spherical representation (as in the baselines of Appendix B.1) – and we select the one with the higher validation MSE (where the validation data is a portion of the data assigned for training).

Fig. 17a shows that in the free-motion benchmark, all the 3 neural models improve the MSE in comparison to the standard KF, yet are outperformed by OKF. Furthermore, while OKF has the best NLL, the more complicated models NKF and NKFH increase the NLL in orders of magnitude. Note that the instability of NKFH is expressed in poor generalization to lower accelerations in addition to the extremely high NLL score.

Fig. 17b shows that in Const_a benchmark, all the 3 neural models improve the MSE in comparison to the standard KF, but only NKFH improves in comparison to OKF as well. On the other hand, NKFH still suffers from very high NLL.

In summary, all 3 variants of NKF outperform the standard KF in both benchmarks in terms of MSE. However, when comparing to OKF instead, aKF and NKF become inferior, and the comparison between NKFH and OKF depends on the selected benchmark and metric.

