# OpenReview forum: "Optimization or Architecture: How to Hack Kalman Filtering"
_NeurIPS.cc/2023/Conference — NeurIPS 2023 poster_

### Official Review · Reviewer_hyJH · 2023-07-04

**Soundness:** 3 good
**Presentation:** 4 excellent
**Contribution:** 2 fair
**Rating:** 6
**Confidence:** 2

**Summary:**

The paper presents a new Kalman Filter (KF) algorithm named Optimized Kalman Filter, which applies an optimization to the supervised learning loss as usually used for training non-linear models (e.g. NKF). For linear KF, a more straightforward way is to estimate the noise covariance, which is directly plugged into the estimator with optimal MSE (if the covariances are known). The authors argue that the significant improvement of NKF is exclusively a result of the optimization approach, as they observe that OKF outperforms NKF in Doppler radar problem, a simple problem with true KF. They also demonstrate the benefits of OKF when the true model is non-linear in Doppler Radar Tracking and Video Tracking problem.

**Strengths:**

The paper demonstrate a simple yet profound neglect in the literature. That is all the previous literature did not fairly compare linear KF and NKF as the two settings differs in both architecture and the way of learning. I am not familiar with the literature of Kalman filtering. However, it is intriguing to see that linear models can performs as well as non-linear ones as long as they are learned properly.

The flow of writing is clear and easy to follow.

The experiments are complete and they cover what it takes to have the main story hold.

**Weaknesses:**

1. For the completeness, it would be nice to have a formal definition of the Neural KF and their optimization goal. How are the models normally set up? Do they also optimize the covariance matrices?
2. The theoretical contribution is limited. All the results are either in the literature or follows immediately from simple algebra. Some interesting theory can be studied. It would be interesting to analyze the error introduced by estimated covariance matrices (Algorithm 1), which reveals their suboptimality. Could we write the closed form of the minimum-variance unbiased estimator when the covariance matrices are not known?
3. In section 4, it would be interesting to compare with KF with known covariance matrices.
4. In section 5, it would be interesting to compare with NKF as well, since the problem is apparently nonlinear

**Questions:**

See the weaknesses discussions. In general, it is not clear whether OKF will still outperform NKF on actually non-linear systems (not very possible when the system is highly nonlinear).

**Limitations:**

The paper did not discuss limitations of their work. I have pointed out a number of directions to improve the theoretical results. The paper conducted experiments on video tracking, which involves predicting human behavior. The authors should discuss the the potential social impact if their algorithm is applied to real products.

---

> ### Author Rebuttal · Authors · 2023-08-07
>
> We thank the reviewer for their helpful feedback. We will modify our manuscript accordingly. Please see our detailed responses below.
>
> **In section 4, it would be interesting to compare with KF with known covariance matrices.**
>
> We conducted this experiment in Appendix B1. The oracle performed very similarly to Algorithm 1 (see Table 2); this indicates that the issue is to choose the right objective, rather than how to optimize/estimate it. Note that an oracle experiment in Section 4 is problematic, since the dynamics are nonlinear and the covariance matrix Q is not well-defined wrt the model F.
>
> **In section 5, it would be interesting to compare with NKF as well, since the problem is apparently nonlinear.**
>
> We indeed tested NKF on our two most nonlinear benchmarks. All our Doppler-radar benchmarks are specified in Table 1 and Figure 9 in the appendix. Section 4 (NKF) corresponds to Fig 9d,9e, which are the most nonlinear and farthest from the KF assumptions. We will make the connection between the benchmarks clear. Thanks for pointing to it.
>
> **It is not clear whether OKF will still outperform NKF on actually non-linear systems (not very possible when the system is highly nonlinear).**
>
> As specified above, we tested NKF vs. OKF on highly nonlinear systems. In fact, as mentioned in Line 20, NKF was the original goal of our work, and we made sure to design a benchmark where the neural network should have an added value.
>
> The failure of NKF might have been caused by many factors: difficulty to learn the nonlinear models, too many degrees of freedom, architecture or optimization choices, etc. Note that the take-away is *not* that neural networks are not good, but rather that their comparison to KF should be done fairly using the *same* optimization. Indeed, in our case, the fair comparison revealed that NKF did not really learn the nonlinear modeling better, a crucial conclusion.
>
> **How are the NKF models normally set up? Do they also optimize the covariance matrices?**
>
> The NKF architecture is reported for completeness in Appendix C and  Figure 14. Its optimization goal is the predictions MSE and optimized parameters are all neural network weights + all covariance parameters. In that sense, you may view OKF as an ablation test of NKF (as mentioned in Line 23).
>
> **Could we write the closed form of the minimum-variance unbiased estimator when the covariance matrices are not known?**
>
> Since our inherent premise is violation of the model assumptions, closed-form solutions are generally not available. Instead, in Proposition 1 and 2 we analyzed the deviation of the optimal parameters from Algorithm 1, for specific violations of Assumption 1. Regarding the effect on the MSE-gap, Table 2 in Appendix B1 indicates that the effect varies greatly between scenarios (OKF is always better, but by different margins). The reviewer’s suggestion of studying these effects theoretically under various assumptions, is indeed a very interesting direction for (probably multiple) future works.
>
> **The paper did not discuss limitations of their work.**
>
> We consider the two main limitations to be: (a) we focus on the supervised setting, as discussed in Lines 53-62; (b) numerical optimization has little theoretical guarantees, as discussed in Lines 155-159. We will clarify the connection of these discussions to the limitations of our work. Thanks for pointing to it.

---

### Official Review · Reviewer_GLsN · 2023-07-06

**Soundness:** 3 good
**Presentation:** 2 fair
**Contribution:** 3 good
**Rating:** 6
**Confidence:** 3

**Summary:**

In the domain of nonlinear filtering, it's a usual practice to draw comparisons between nonlinear architectures like neural networks and traditional linear Kalman Filters (KF). This approach blends two distinct aspects: the nonlinear structure and the numerical optimization technique, thereby introducing potential issues in the comparison process. Specifically, nonlinear models are often subjected to optimization, unlike the benchmark KF models.

This paper proposes a solution to these challenges by introducing the Optimized KF (OKF), a method designed to customize numerical optimization in alignment with positive-definite KF parameters. The authors demonstrate that the apparent advantage of neural networks over KF can be completely neutralized when KF is optimized using OKF. Furthermore, our tests reveal that OKF surpasses the standard KF in various problem situations, suggesting that KF might present a sub-optimal choice.

The key contributions of this paper include:

1. Highlighting a prevalent methodological error of comparing an optimized filtering model against an unoptimized KF.
2. Introducing the Optimized KF (OKF) as a solution, showcasing the experimental results in comparison with nonlinear structures like existing neural networks and conventional linear KF, after switching the benchmark to OKF and utilizing the modified overall experimental outcomes.
3. Establishing that OKF consistently achieves superior accuracies compared to KF across a spectrum of problems.
4. Offering a theoretical analysis that highlights the sub-optimality of KF.

**Strengths:**

1. Motivation is clear and it seems like a meaningful study.
2. It is well-proven formally.
3. The proposed model seems novel.

**Weaknesses:**

The idea in this paper is very novel, but it feels like it just added gradient based optimization at the end.
Is there any comparison with nonlinear architecture (like Neural network)?

**Questions:**

1. Is the optimization method only using gradeient descent?
2.  Is there any comparison with nonlinear architecture (like Neural network)?

**Limitations:**

The idea in this paper is very novel, but it feels like it just added gradient based optimization at the end. It would be nice to suggest various optimization methods or optimization methods suitable for Kalman Filter.

---

> ### Author Rebuttal · Authors · 2023-08-07
>
> We thank the reviewer for their helpful feedback. Please see our responses below.
>
> **It would be nice to suggest various optimization methods.**
>
> Our key insight is that noise estimation (Algorithm 1) is not a reliable proxy to MSE optimization, and instead the MSE should be optimized directly. The focus is on choosing this “right” objective to optimize; choosing how to optimize it, between SGD, Adam, etc., is beyond our scope.
>
> **Is the optimization method only using gradient descent?**
>
> Mostly, but not only: as discussed in Section 3, we also overcame the challenge of optimizing the parameters without breaking their symmetric and positive-definite structure. For this, we used the Cholesky parameterization - a trick that can be applied on top of existing optimization methods.
>
> Still, how to numerically-optimize the MSE is not our main novelty. As discussed above, the main novelty is choosing to optimize the MSE at all, instead of the proxy task of noise estimation, after we observe that the two are usually not equivalent.
>
> **Is there any comparison with nonlinear architecture (like Neural network)?**
>
> Section 4 presents a comparison between the optimized linear KF (OKF) and the nonlinear neural network (NKF). Appendix C also experiments with other neural network variants.

---

### Official Review · Reviewer_mJXo · 2023-07-08

**Soundness:** 2 fair
**Presentation:** 2 fair
**Contribution:** 3 good
**Rating:** 5
**Confidence:** 3

**Summary:**

The paper is considering a linear Gaussian filtering problem where the noise covariance are unknown. It assumes it has access to state and observation data. It aims to find the noise covariances by minimizing the estimation error of a Kalman Filter using those noise covariance  matrices. This is in contrast to the standard fitting approach of Algorithm 1. The empirical results show support for this approach compared to algorithm 1

**Strengths:**

- The paper has an honest tone that I appreciate
- the idea of using the filtering error to optimize the filtering parameters is natural and reasonable.

**Weaknesses:**

- It is very difficult to follow the paper from the beginning and the title. It seems that this is continuation of the author's research, but the readers do not have the same context.
- It is not exactly clear what is the main message of the paper. Is the paper arguing against using neural networks for nonlinear filtering problems?
- There are also other, more filtering oriented, ways to estimate the covariance matrices, like optimizing Q and R to make the innovation error white. This is in the references in the paper, but [1] is also recent review of the topic. It will be good to do comparison with them
- There are no guarantees for solving problem (3), but algorithm (1) has guarantees.

[1] Zhang et. al. On the Identification of Noise Covariances and Adaptive Kalman Filtering: A New Look at a 50 Year-Old Problem

**Questions:**

Please see the weakness above

---

> ### Author Rebuttal · Authors · 2023-08-07
>
> We thank the reviewer for reading our work and providing honest feedback. We respond to the reviewer’s comments below. More importantly, the review specifies that the main message was not clear to the reviewer. **Since it is important for us to make the presentation as clear as possible, we recap our main message here in different words, and attach a new explanatory diagram (attached in the global rebuttal post). Does the diagram help? We will appreciate any suggestions regarding which points are not clear and why.** We will also appreciate it if the reviewer could re-evaluate our contribution accordingly.
>
> ### Our recapped message
> 1. The KF assumptions often do not hold.
> 2. In such cases, instead of viewing Q,R as representatives of the noise, we should view them as parameters of the prediction model.
> 3. Accordingly, we should optimize Q,R wrt the ultimate objective (MSE) rather than the proxy objective (noise estimation), since the proxy is no longer equivalent to the ultimate goal.
> 4. In particular, when comparing KF to a neural network, notice that the neural network is indeed optimized for the MSE; for the comparison to be fair, the KF must be optimized similarly.
> 5. We show that the literature is generally unaware of (4), and demonstrate that by not optimizing the KF, the whole experimental conclusion may be reversed.
> 6. We suggest how to optimize the KF in a simple way despite the SPD structure of the parameters Q,R.
>
> ______________________________
>
> ## Responses to specific questions
>
> **Is the paper arguing against using neural networks for nonlinear filtering problems?**
>
> Certainly not. Neural networks may perform better or worse than the KF. Our claim is that to decide between these two possibilities, one must test the network and the KF under the same conditions, including similar optimization of their parameters. In Section 4 we demonstrate that this methodological issue is crucial: NKF outperforms the KF if the KF is not optimized, but not after the KF is optimized (by OKF).
>
> **There are also other, more filtering oriented, ways to estimate the covariance matrices… [1] is also recent review of the topic. It will be good to do comparison with them.**
>
> As discussed in Lines 53-62, we address the supervised setting, where both states {x} and observations {z} are available for training, and thus the simple Algorithm 1 can be applied. In different settings, when the states {x} are unknown and Algorithm 1 is not applicable, many studies such as [1] propose alternative methods to estimate the noise.
>
> Our main claim is that the objective itself - noise estimation - should often be replaced. To stress this, we compare to an *oracle* baseline that knows the *exact* noise Q,R (Appendix B1). Contrary to the oracle, [1] operates in *harder* conditions than Algorithm 1, and thus using it as a baseline will not strengthen our message. However, we will stress the additional oracle baseline in the front manuscript. Thanks for pointing to this issue.
>
> **There are no guarantees for solving problem (3), but algorithm (1) has guarantees.**
>
> Our main premise is that the KF assumptions often do not hold, and their violation may be tricky to even notice (Lines 217-243). Hence, as discussed in Line 155, neither Algorithm 1 nor Algorithm 2 actually have optimality guarantees. However, Algorithm 2 is at least aligned with the desired MSE objective, whereas Algorithm 1 is not. For example, as demonstrated in Appendix B2, instead of converging to optimal predictions, Algorithm 1 may in fact *deteriorate* in MSE when fed with more data.

---

> > ### Comment · Reviewer_mJXo · 2023-08-15
> >
> > Thanks for the response and re-expressing your main message in a simpler way. I keep my score because the message is not strong. In a linear-Gaussian setup, the KF assumptions holds, so it makes sense to identify the noise matrices separately if one has access to data from the state. In a nonlinear setup, the KF update equations does not make sense, and it is better to just optimize for a constant gain (instead of indirectly compute it from learned Q and R), or use alternative approaches like EnKF that rely on a simulator for the state (which the paper kind of assumes when access to data from the state is assumed). The NKF approach is not as common in the nonlinear filtering, because even if it is optimized correctly, it does not give an exact solution.

---

> > > ### Author Response · Authors · 2023-08-15
> > >
> > > Dear Reviewer mJXo,
> > >
> > > We beg to differ concerning the strength of the message. There are at least tens of papers that make the mistake of comparing optimized neural networks to non-optimized KFs. We listed 10 of them in Lines 33-37. Before reading our paper, were you aware of this “profound neglect in the literature”, as put by Reviewer hyJH?
> > >
> > > The linear Gaussian case is, as said by David Thomson, a fairytale we tell our students; it rarely holds. Still, the KF is one of the most popular algorithms in the world, and its equations are not considered to “not make sense”, as put by the reviewer, despite the non-linearities in the world.
> > >
> > > Our message implies that much of the non-linear filtering literature is wrong, and that much of the linear filtering practice can be improved. We feel it is a strong message. Wouldn't you agree?
> > >
> > > Minor:
> > > * **”The NKF approach is not as common…”**: We cite a sample of 10 such papers in Lines 33-37.
> > > * **”...because even if it is optimized correctly, it does not give an exact solution”**: Neural network models in general learn approximated solutions rather than exact ones, and are still popular and successful in many fields, including filtering as mentioned above.
> > > * **”use alternative approaches like EnKF”**: In addition to the importance of the KF by itself, please note that our method’s benefits are also demonstrated on top of a non-linear alternative, namely, the EKF (Appendix B1).

---

> > > > ### Comment · Reviewer_mJXo · 2023-08-20
> > > >
> > > > Thanks for the comment and apologies for my late response.
> > > >
> > > > 1- I do not agree with the article’s characterization of the nonlinear filtering literature. If the paper is considering a nonlinear filtering problem, then the baseline is Monte Carlo based approaches like ensemble Kalman filter (which is different than extended Kalman filter) and SIR particle filters, not Kalman filter. This is not a common miss conception in the majority of nonlinear filtering literature. It is definitely a profound neglect in these 10 papers, but not in the majority of the nonlinear filtering. Please see the textbook
> > > >
> > > > Data Assimilation: A Mathematical Introduction. Andrew M. Stuart, Kody Law, and Konstantinos Zygalakis
> > > >
> > > > 2- As a result I do not find the optimized Kalman filter proposed in the paper the right baseline for a nonlinear filtering problem. For a linear problem with known dynamic and observation matrices, but incorrect noise covariance, it is best to identify the noise covariance matrices with the state trajectory data. In a nonlinear setup, the equation of Kamlan filter do not hold, so one might as well optimize directly for the Kalman gain instead indirectly computing it from noise matrices.
> > > >
> > > > 3- Finally, my last point was about criticizing the neural net based approach because even with infinite capacity and accurate optimization, it can not provide the optimal estimate. It is necessary to compute the posterior in a nonlinear setup.
> > > >
> > > > 4- My conclusion from the results presented in the paper is that in several nonlinear filtering problems, a filter with linear architecture is already good. Which is somewhat known due to success of Ensemble Kalman filter.
> > > >
> > > > The current message of the paper is still not convincing to me. This is probably due to the difference in background and difference in exposure to the literature. I understand the other reviewers found the message compelling, so I will increase my score. But I can not give higher score due to the points I raised above. I hope you will find the review useful in any case.

---

> > > > > ### Author Response · Authors · 2023-08-20
> > > > >
> > > > > Thank you very much for presenting your concerns and perspective in detail, which will help us enhancing the discussion of our limitations. Our responses are below.
> > > > >
> > > > > 1. We are aware of the many different approaches for nonlinear filtering. We believe this only increases the importance of the insight, that the standard KF may be competitive with nonlinear methods, once it is tuned correctly.
> > > > >
> > > > > 2. Optimizing (Q & Kalman gain) instead of optimizing (Q & R) is an interesting idea for representation shift, and can be studied in future work, but it does not affect our message or contribution.
> > > > >
> > > > > 3. In realistic filtering problems, no exact solution can be found for the unknown non-linear dynamics, and neural networks are a valid approach. Note that our neural model NKF does calculate a posterior.
> > > > >
> > > > > 4. We conclude not only that a linear architecture can be competitive, but also - more importantly - that this competitiveness can be easily missed when the linear model is not optimized.

---

> > > > > > ### Comment · Reviewer_mJXo · 2023-08-20
> > > > > >
> > > > > > A clarification, in point number 2, I meant combining the prediction and correction step of the Kalman filter and learning the combined gain ( so it only needs learning one gain matrix)
> > > > > >
> > > > > > Also, what is the difference of OKF with tuning the Kalman filter, as explained in this Matlab tutorial?
> > > > > >
> > > > > > https://www.mathworks.com/help/fusion/ug/tuning-kalman-filter-to-improve-state-estimation.html
> > > > > >
> > > > > > 3- Particle filters give exact solution in the limit as the number of particles goes to infinity. I think it is important to look for algorithms that are at least asymptotically exact!

---

> > > > > > > ### Author Response · Authors · 2023-08-21
> > > > > > >
> > > > > > > **Separating Q and R** may still be meaningful when the assumptions do not hold, e.g., to calculate posteriors both before and after the measurement (in appendix C, for example, we use posteriors to measure NLL). Q,R also carry semantic meaning (as discussed after Proposition 1). Yet, we agree that in some scenarios, simplifying to a single-step model is natural in the optimization framework. We will add this to the discussion. Thanks for pointing to it.
> > > > > > >
> > > > > > > **Regarding the Matlab tutorial**: As we mention in Lines 34 and 347-348, certain works replace noise estimation with a heuristic tuning of the parameters, such as trial-and-error. The Matlab tutorial suggests another kind of heuristic, where the 2x2 covariance matrix is replaced with a scalar. By contrast, we address the full SPD covariance matrix (not just scalar or diagonal), which permits full expressiveness. We demonstrate that this is crucial - both in Section 5.1 after Proposition 1, and in the new experiments from the rebuttal with Reviewer JSXK.
> > > > > > >
> > > > > > > Finally, while the possibility to optimize is known to a certain point, the *need* to optimize is less understood. Optimization is typically seen as a fallback if direct estimation is not feasible. This is also implied in the Matlab tutorial: they first propose direct variance estimation, and then write "Sometimes measurement parameters such as measurement noise may not be known", and move to the heuristic scalar optimization. Maybe this view comes from the lack of general, fully-expressive, end-to-end optimization methods for KF. We propose such a method (OKF), and analyze why it is crucial to prefer it over noise estimation - both theoretically and empirically.

---

> > > > > > > > ### Comment · Reviewer_mJXo · 2023-08-21
> > > > > > > >
> > > > > > > > Thanks for explaining the comparison to the Matlab tutorial. My point was to show that optimizing the noise covariance to minimize the error, which is referred to as tuning, is not a profound neglect in nonlinear filtering. The tutorial describes automating the procedure by optimization. It seems that the papers cited in your article did not tune the KF correctly.

---

### Official Review · Reviewer_JSXK · 2023-07-09

**Soundness:** 3 good
**Presentation:** 3 good
**Contribution:** 3 good
**Rating:** 6
**Confidence:** 3

**Summary:**

This paper uses stochastic optimization to estimate the covariance of Kalman filters in supervised tracking problems. They find that online stochastic optimization in this manner tends to generalize better than either neural Kalman filters with LSTM base models or the older plug-in estimators for Kalman filter problems. Their argument for the outperformance of stochastic optimization in these problems relies heavily on the fact that the assumptions for the plug-in estimators ends up being violated in some manner, producing at least some amount of model mis-specification. The experiments cover a bunch of simulated data where the assumptions of the non-optimized Kalman filter can be tested rigorously, as well as less toy video tracking and lidar state estimation experiments.

After the authors rebuttal I'm a bit happier about this paper. Thanks for showing the value of the Cholesky parameterization and answering the rest of my questions.

**Strengths:**

Originality

-	I don’t have a great background in the Kalman filtering literature so I find it difficult to fairly judge originality. However, based on a cursory reading of the other literature, it seems to be a quite original work.

Quality

-	Overall the paper seems to be of pretty high quality experiment-wise and technically. See typos at bottom.

Clarity

-	The assumptions are spelled out pretty well, especially as each experiment tends to have a different set of assumptions.

Significance

-	This paper ends up being yet another demonstration of “stochastic optimization tends to work better in practice than in theory” which seems certainly like a good addition to the ML community, even if it comes from a place that is a bit outside.

Indeed, appendix b.2 shows that stochastic optimization is a bit more robust to model mis-specification in what you describe, as you show an example where adding more data definitely makes the error worse. Honestly, you could move this into the main text with some tigher writing.

-	Overall, between the authors’ software and the simplicity of the approach, it also seems like a fairly reasonable contribution to the system identification literature.

**Weaknesses:**

Originality

-	From my limited knowledge here, I do think that the Bayesians have been estimating parameters in a similar manner for GPs in state identification for a while. See for example [1], [2]

Quality

-	A lack of comparison to non-Kalman filter benchmarks on some of these tasks limits the relevance to the ML community – perhaps physics NNs or even a direct LSTM would be a good baseline.


Clarity

-	I think that I am a little confused as to the broader application of the Kalman filter in this type of supervised learning setting however. The motivation for this specific subset (supervised like training estimation) could be explained a bit better.

-	The implications of Assumption 1 should probably be explained a bit better right after the assumption and theorem is sketched out. Theorem 1 is also not really well written or formally stated – please either write it more formally or just drop the theorem heading.

Significance

-	Touched on a bit above, but I am a little puzzled by the when is the supervised estimation Kalman filtering problem actually used? From what I can tell all of the experiments are very distinguished from reality and simplified.

-	At the end of the day, the approach is really just covariance matrix estimation for trajectories, which seems like it has been done before in things like [3], [4]

References:

[1] Reece and Roberts, 2010. An Introduction to Gaussian Processes for the Kalman Filter Expert

[2] Deisenroth, Huber, & Hanebeck, ICML, 2009. Analytic moment-based Gaussian process filtering.

[3] Xiao and Wu, Ann. Stat, 2012. Covariance matrix estimation for stationary time series.

[4] Wang et al, Neurocomputing, 2017. An adaptive Kalman filter estimating noise covariance.

**Questions:**

When Assumption 1 is not violated, does the stochastic optimization approach converge to the same solution (just at a slower rate)?

You may wish to break Assumption 1 into four separate assumptions – stationarity, i.id-ness, known covariances, and good initial states.
This would probably cleanup some of the writing in the experiments by being able to which specific piece of assumption 1.

Could the neural Kalman filter be combined with the optimized Kalman filter with both being optimized jointly?

Could the authors run an ablation where the L Cholesky factor is parameterized diagonally? What is the impact of the full parameterization?
I believe that this should become much less well-specified and harder to optimize.

Is the formulated stochastic optimization problem still convex or at least reachable to a global optimum?

Why can we not map observations from cartesian to polar coordinates in the lidar and doppler experiments, thereby producing observations that are much less violating of assumption 1.

Section 4:
In the doppler radar problem, could another benchmark (either a physics neural net or a neural ode) be used for the trajectory problem as well?

typos / writing:

L357: studied -> studies

Section 5: use a real name for section heading – e.g. “Comparing Optimized KF to KF”
Personally, I would probably combine sections 4 and 5 into a single section with the same sub-headings: experiments. Currently, the headings seem a bit informal.

**Limitations:**

I think motion tracking in crowds may not be a great application socially, but otherwise no other societal impacts.

---

> ### Author Rebuttal · Authors · 2023-08-07
>
> We thank the reviewer for their detailed feedback. **Before responding in detail and presenting the new experiments as requested, we wish to focus on the comments about the significance of our contribution. This may clarify some of the reviewer’s questions, and we will appreciate it if the reviewer could re-evaluate our contribution given this discussion.**
>
> ## Discussion on contribution
>
> **”When is the supervised estimation Kalman filtering problem actually used?”**
>
> As mentioned in Line 20, our work in this setting originated from an actual real-world radar problem. The supervision of true states (or more precisely, states data with negligible errors) used to come from either monitored experiments or simulations. Following the review, we will discuss this more clearly in the scope paragraph. In addition, note that Lines 55-57 provide few other brief motivational examples. Finally, all the 10 works cited in Lines 33-37 follow this supervised filtering setting, indicating its relevance.
>
> **MSE optimization $\ne$ covariance noise estimation**
>
> We do not attempt to estimate the noise covariance matrix better than others; such estimation is trivial in our setting (by Eq. 2). The extensive literature on this topic corresponds to *other* settings (e.g., when the states {x} are unavailable in the train data). Instead, we observe that this common noise-estimation task is usually *not* a proxy to MSE optimization, hence we should often *give up* on accurate noise-estimation in favor of direct MSE optimization. As the reviewer mentioned, Appendix B2 shows that these goals are *contradictory*. Hence, **our key contribution is not *how* to optimize, but rather *what* to optimize** (although we also overcome how to optimize SPD parameters). We tried to convey this message in the intro. **Following the rebuttal, we will specify this distinction more clearly, and will add an explanatory diagram (which is attached in the global rebuttal post)**.
>
> _______
>
> ## Detailed responses
>
> As suggested, we will address the typos and the presentation of the preliminary assumption and theorem. Thanks for pointing them out.
>
> **Could the authors run an ablation where the L Cholesky factor is parameterized diagonally?**
>
> **We now ran it** on all the 5 Doppler benchmarks of Table 1 in Appendix B1. We added *DKF* - a variant of OKF with Diagonal parameterization of Q and R. On the first 3 benchmarks, where the target motion is linear, DKF was indistinguishable from OKF. However, on the 2 non-linear benchmarks (the same benchmarks used in Section 4), DKF was inferior to OKF (but superior to KF).
>
> | Benchmark   | KF    | DKF   | OKF   |
> |-------------|-------|-------|-------|
> | Toy  | 151.7 | 84.2  | 84.2  |
> | Close | 25.0  | 24.8  | 24.8  |
> | Const_v  | 90.2  | 90.1  | 90.0  |
> | Const_a ("no turns") | 107.5 | 106.1 | 101.6 |
> | Free motion | 125.9 | 121.1 | 118.8 |
>
> **When Assumption 1 is not violated, does the stochastic optimization converge to the same solution (just at a slower rate)?**
>
> We typically use Algorithm 1 to initialize $Q,R$ for Algorithm 2. Under Assumption 1, this initialization is the globally-optimal solution. Thus, the GD in Algorithm 2 (with a sufficiently small learning rate) will just remain in this optimum. **We now added an empirical demonstration of this phenomenon**, on a toy version of the lidar problem, which satisfies Assumption 1. As expected, throughout the entire training process, OKF’s validation MSE remained identical to KF’s MSE.
>
> **Could NKF be combined with the optimized KF with both being optimized jointly?**
>
> The NKF already includes optimization of Q and R, i.e., already combines OKF’s optimization of Q,R with the neural architecture.
>
> **Is the formulated stochastic optimization problem still convex?**
>
> Since our main premise is violation of the model assumptions, there are generally no theoretical guarantees of convexity. Instead, Algorithm 2 is backed by (a) empirical evidence in Sections 4,5; (b) external evidence of the strength of stochastic optimization in general; (c) the motivation to optimize the MSE rather than the contradicting objective of Algorithm 1.
>
> **Why can we not map observations from cartesian to polar coordinates, thereby producing observations that are much less violating assumption 1.**
>
> Following this motivation exactly, we produced the simplified toy environment where the noise is *Cartesian* i.i.d (Section 5.1). However, in the non-toy problem, the discrepancy is inherent and not just a matter of representation: the MSE objective is Cartesian, whereas the radar noise is polar.
>
> **A lack of comparison to non-Kalman filter benchmarks on some of these tasks limits the relevance to the ML community.**
>
> The relevance to the ML community comes from two points:
> 1. An *application of ML* to real-world filtering: thanks to the powerful ML optimization tools, we can easily optimize the MSE, instead of the wrong proxy objective of noise estimation.
> 2. Experimental methodology when testing new models against classical ones: both models should be tested under the same optimization method.
>
> We support both points in a variety of experiments in Sections 4, 5.1, 5.2, 5.3, B.1, B.2, B.3, C.
> However, we did not consider additional neural baselines (in addition to the 3 neural KFs in Appendix C) as supportive of our claim - since we do *not* claim that OKF is in general superior to neural networks (see Lines 36, 204).
>
> **”Bayesians have been estimating parameters in a similar manner for GPs in state identification.” / “The approach is really just covariance matrix estimation for trajectories, which seems like it has been done before in things like [3], [4].”**
>
> Please see the discussion above about our contribution: covariance matrix estimation has indeed been studied extensively; however, we argue that covariance estimation is often not the desired objective, and instead propose to tune the KF parameters via the *contradicting* MSE objective.

---

> > ### Comment · Reviewer_JSXK · 2023-08-20
> > **thanks for the response**
> >
> > Sorry for the late reply here and thank you for the response and study on the diagonal Cholesky factor. I've raised my score to a 6.
> >
> > However, one final question here: when you state that "The NKF already includes optimization of Q and R, i.e.,.." does this imply that the NKF is then a strictly more general method of the OKF? In which case, its underperformance is driven by optimization difficulties (e.g. overfitting) rather than being the wrong approach?

---

> > > ### Author Response · Authors · 2023-08-20
> > >
> > > Thank you for your response.
> > >
> > > Regarding the question, you are right: NKF is more general than OKF. In fact, we originally implemented OKF as an ablation test for NKF (as mentioned in Line 23), and the advantage of OKF came to us as a surprise. There is nothing wrong with the approach of NKF (and we will make sure to stress it in the paper). In fact, its failure against OKF only motivated us to try improving NKF (some of these attempts are reported in Appendix C). Even if we succeeded improving NKF, the OKF reference would still play a crucial role, since it indicated there might be room for improvement. In this hypothetical case, our message would still hold: neural networks are a valid tool for filtering, but to be evaluated, they should be compared to OKF rather than the non-optimized KF.

---

### Author Rebuttal · Authors · 2023-08-07

We thank all the reviewers for their helpful comments. We are pleased that the reviewers found our work “**original**” (JSXK), **”honest”** (mJXo), **”meaningful”** and “**very novel**” (GLsN), and acknowledged that we point to a **“prevalent methodological error”** (GLsN) and a **“profound neglect in the literature”** (hyJH). The reviews considered our work **“high quality technically”** (JSXK) and **”well-proven formally”** (GLsN). They found our experiments “**complete**” (hyJH), covering a “**spectrum of problems**” (GLsN), each with a **“different set of assumptions”** (JSXK).
Finally, they considered our motivation **“clear”** (GLsN) and writing **”clear and easy to follow”** (hyJH).

Below we respond to the reviews and discuss the consequent modifications in our manuscript. We also include two additional experiments following JSXK’s suggestions, and attach a new diagram that summarizes our main message.

---

### Decision · Program_Chairs · 2023-09-21

**Decision:**

Accept (poster)

**Comment:**

This paper's primary contribution is to point out that online adaptation of the linear Kalman filter drastically improves its performance, even in nonlinear dynamical systems, outperforming neural baselines. Reviewers generally agreed that the experiments were high-quality, and the message was well-communicated; on the other hand, they agreed that the novelty of the other contributions (new algorithms, and theoretical understanding) was limited. There was also some divergence of opinions on whether this message was sufficiently novel.

Thus, the paper decision hinges upon the subjective question of whether pointing out this (quoting a reviewer) "neglect in the literature" is a standalone contribution. After discussion and evaluating the other reviews, although consensus was not 100% reached, the reviewer with the strongest objection adjusted their score slightly. Overall, there seems to be enough precedent for the prevalence of non-adapted Kalman filters in experiments, to bring this paper's value above the acceptance threshold.

The reviewers have pointed out a couple of things that would immediately make the paper more convincing: a comparison with the ensemble KF, and (any) attempt at theoretical understanding. The authors are encouraged to incorporate these comments.